# High-frequency head impact causes chronic synaptic adaptation and long-term cognitive impairment in mice

Stephanie S. Sloley[1,2,5], Bevan S. Main[2,5], Charisse N. Winston[1,2], Alex C. Harvey[2], Alice Kaganovich[3], Holly T. Korthas[1,2], Adam P. Caccavano[1,4], David N. Zapple[2], Jian-young Wu[1,2], John G. Partridge[1,4], Mark R. Cookson [3], Stefano Vicini[1,4] & Mark P. Burns [1,2✉]

Repeated head impact exposure can cause memory and behavioral impairments. Here, we report that exposure to non-damaging, but high frequency, head impacts can alter brain function in mice through synaptic adaptation. High frequency head impact mice develop chronic cognitive impairments in the absence of traditional brain trauma pathology, and transcriptomic profiling of mouse and human chronic traumatic encephalopathy brain reveal that synapses are strongly affected by head impact. Electrophysiological analysis shows that high frequency head impacts cause chronic modification of the AMPA/NMDA ratio in neurons that underlie the changes to cognition. To demonstrate that synaptic adaptation is caused by head impact-induced glutamate release, we pretreated mice with memantine prior to head impact. Memantine prevents the development of the key transcriptomic and electrophysiological signatures of high frequency head impact, and averts cognitive dysfunction. These data reveal synapses as a target of high frequency head impact in human and mouse brain, and that this physiological adaptation in response to head impact is sufficient to induce chronic cognitive impairment in mice.

[1] Interdisciplinary Program in Neuroscience, Georgetown University Medical Center, Washington, DC, USA. [2] Department of Neuroscience, Georgetown University Medical Center, Washington, DC, USA. [3] Laboratory of Neurogenetics, National Institute of Aging, Bethesda, MD, USA. [4] Department of Pharmacology and Physiology, Georgetown University Medical Center, Washington, DC, USA. [5]These authors contributed equally: Stephanie S. Sloley, Bevan S. Main. ✉email: mark.burns@georgetown.edu

It is difficult to disassociate the effects of brain trauma pathology from the physiological events that occur in the brain after exposure to head impacts. While the neuropathology of the neurodegenerative disease associated with repeated head impact, chronic traumatic encephalopathy (CTE), has been well described[1–3], little is known about the effect of head impacts on brain physiology. In high-school football players, changes in brain function are apparent in players after exposure to non-concussive head impacts[4], indicating that impact-induced changes in brain physiology could be occurring in the absence of structural damage and accumulate as the frequency of head impacts increase[4]. Preclinical studies have classically associated cognitive deficits in mild traumatic brain injury (mTBI) models to axonal injury, inflammation, and p-tau pathology[5–11]. More recent models of repeat mTBI with low levels of pathology still present with cognitive dysfunction, suggesting an alternative mechanism of action in the development of cognitive impairment that is independent of pathological changes[12–16]. Such mechanisms, however, remain poorly defined at the molecular or cellular level.

The average college football player receives 21 head impacts per week, with defensive ends receiving 41 head impacts per week[17]. To study the physiological changes that occur following head impact, we developed a mouse model of very mild impact that mimics this large number of head impacts in a single week. In the current study, we utilized a high-frequency head impact (HF-HI) paradigm, with 30 head impacts in 1 week to better model the human exposures that occur during a week of contact sport[17]. We give these head impacts consecutively, as many of the head impacts can be received in an individual play when a player is tackled to the ground. We have previously reported that 30 head impacts delivered once a day over 6 weeks (low-frequency head impact) does not produce cognitive dysfunction[14], does not produce p-tau or Aβ accumulation, and presents with no evidence of inflammation, cell death, or axonal damage outside the optic tract pathway. Here, we report that exposure to a high frequency of head impacts leads to chronic cognitive deficits in the absence of key TBI or CTE pathology and we determine that these behavioral deficits are due to chronic adaptation of the excitatory synapse. Collectively, these data nominate synaptic adaptations as a potential mechanism for early abnormal behavioral events observed after repeated head impact and suggest that impacts below that required to generate neuropathological sequelae may still cause substantial neuronal dysfunction when delivered at a sufficient frequency.

## Results

**HF-HI causes modifications to the synaptic transcriptome at 24 h post impact.** Mice were exposed to HF-HI before undergoing an acute assessment of working memory at 24 h after the final impact. The T-maze is a test of spontaneous alternation that is dependent on the integrity of limbic and non-limbic pathways, including the basal forebrain, the hippocampus, and the prefrontal cortex[18]. We found that HF-HI mice had significantly fewer spontaneous alternations than sham mice indicating impaired working memory (Fig. 1b; $P = 0.048$). To nominate candidate causative factors for cognitive impairment, we performed RNA sequencing (RNA-seq) to generate an unbiased high-resolution transcriptomic profile of HF-HI. Given that axonal injury, neurovascular damage, and inflammation have all been implicated in CTE, we used whole cortical homogenates to avoid excluding any cell type that could contribute to the observed behavioral phenotype. Hierarchical clustering of the aligned and normalized read counts indicated that our two treatments (sham and HF-HI) clearly separated in terms of overall cortical gene expression (Fig. 1c and Supplementary

Fig. 1B) and this was confirmed with a principal component analysis of the altered gene expression profile (Fig. 1d). This separated profile consisted of 3836 significantly altered genes, with 1956 genes lower and 1880 genes higher expressed in HF-HI compared to sham (Fig. 1e). To gain insights into the molecular mechanisms and pathways associated with gene expression responses, we performed gene ontology (GO) enrichment. Analysis of biological process (BP) categories revealed multiple synaptic signatures, including synaptic signaling (GO:0099536), anterograde synaptic signaling (GO:0098916), and cell–cell signaling (GO:0007267) groups (Supplementary Fig. 1C). Analysis of cellular compartment (CC) categories revealed neuronal components as the most represented feature, with neuron part (GO:0097458) and multiple synaptic modalities, including synapse (GO:0045202), post synapse (GO:0098794), and synaptic membrane (GO:0097060) highly represented (Supplementary Fig. 1C). Directionality analysis showed that the majority of these pathways were not simply upregulated or downregulated, but instead were modified such that the synaptic pathways and CCs appeared in both the significantly upregulated and significantly downregulated pathways (Fig. 1f). Gene analysis revealed a robust up- and downregulation of individual genes involved in the synaptic compartment of neurons, specifically the glutamatergic synapse (Supplementary Fig. 1D, E), and to genes involved in synaptic signaling (Fig. 1g) and long-term potentiation (LTP) (Fig. 1f). To ensure robustness of the dataset, we used real-time quantitative PCR (RT-QPCR) for technical validation of a subset of significantly upregulated genes from the synaptic signaling category (Supplementary Fig. 1E). To study the regionality of this effect, we also confirmed that a subset of these synaptic genes was also altered in the hippocampus of HF-HI mice at 24 h (Supplementary Fig. 1F).

To gain further mechanistic insight into how these altered genes and ontology pathways connect across multiple levels, false discovery rate (FDR) cytoscape enrichment analysis was performed to aid the identification and visualization of connected networks, which contain similar biological themes across several categories. We found significant enrichment in multiple synaptic categories, including receptor internalizations and postsynaptic receptor localization, as well as features of the excitatory synapse such as LTP and plasticity-related categories. In addition, smaller networks describing genetic changes related to neurodegenerative disorders and circadian alterations were also identified by Kyoto encyclopedia of genes and genomes (KEGG) analysis (Supplementary Fig. 2).

Collectively, these results show that there are robust acute transcriptional changes in response to HF-HI that are enriched for synaptic terms, implying a primarily neuronal contribution to the acute behavioral phenotype of this model.

**HF-HI does not cause microglial activation or accumulation of p-tau or Aβ40.** To study neuroinflammation in HF-HI mice, we analyzed Iba1 and CD68 immunohistochemistry in the mouse brain. We found no effect of HF-HI on microglia/macrophage cell number or % area coverage in the hippocampus or corpus callosum (Supplementary Fig. 3A), but did observe a significant effect of HF-HI on microglia/macrophages in the optic tract at 1 day and 1 month post impact ($F_{(3,17)} = 23$, $P < 0.0001$) (Supplementary Fig. 3A). This optic tract-specific inflammation is also seen in our low-frequency head impact model (which occurs without cognitive deficits)[14], and other mouse models of repetitive mTBI and blast TBI[16,19,20], showing that the optic nerve and tract is particularly vulnerable to head impact in rodent models. To further explore inflammation, we analyzed levels of the inflammatory cytokines interleukin-1β and tumor necrosis factor-α in both cortex and hippocampal homogenates at 24 h and 1 month, and

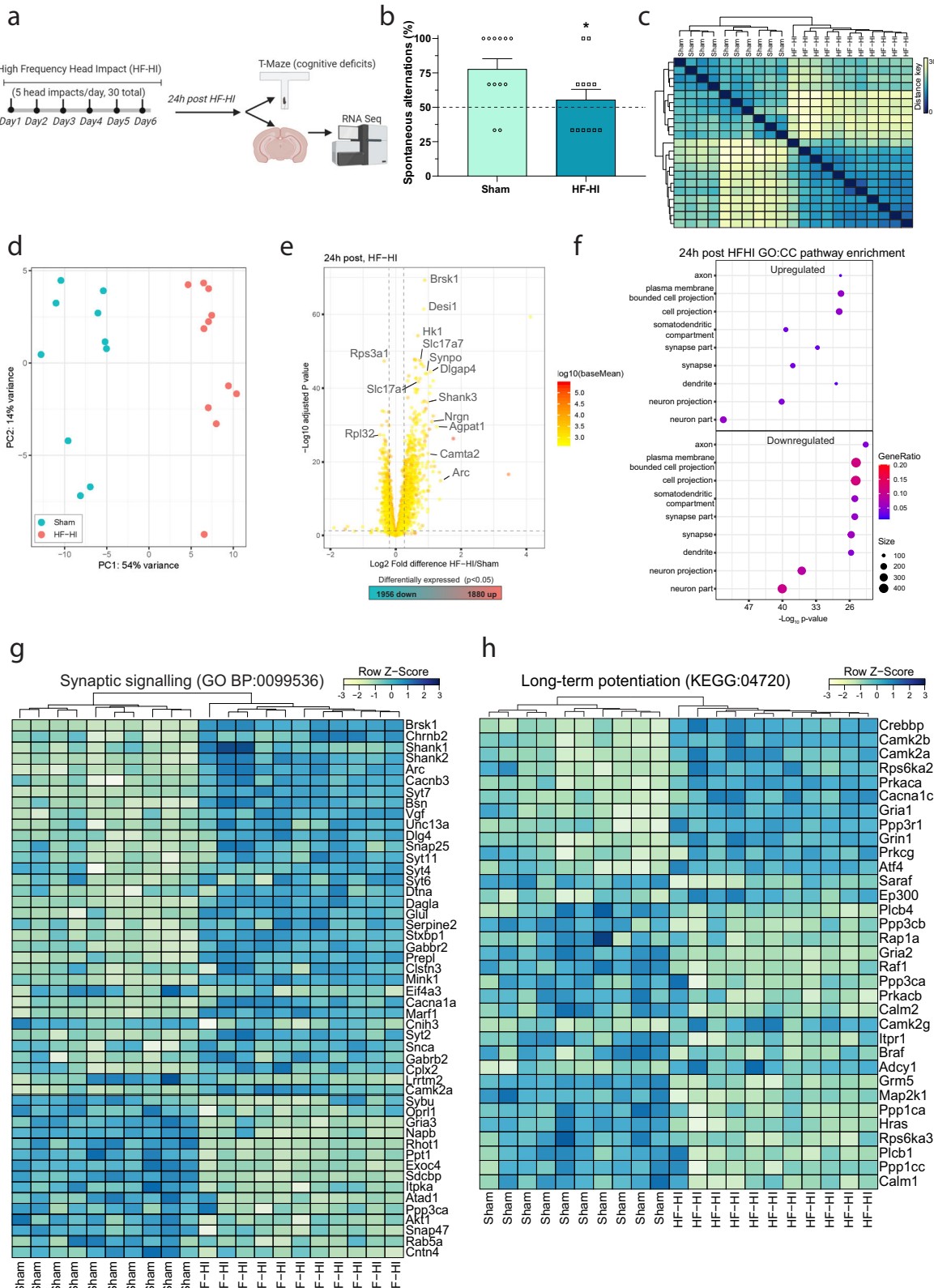

found no effect of HF-HI (Supplementary Fig. 3B, C). We also found no change in *Gfap* or *Aif1* messenger RNA (mRNA) levels after HF-HI (Supplementary Fig. 3D).

We next determined if HF-HI altered $A\beta_{40}$ or p-tau levels. Western blot analyses for multiple phosphorylated tau epitopes revealed no differences between HF-HI and sham animals in the 24 h cortex (Supplementary Fig. 4A) or hippocampus

(Supplementary Fig. 4B), or the 1 month cortex (Supplementary Fig. 4E) or hippocampus (Supplementary Fig. 4F). HF-HI did not change $A\beta_{40}$ levels in the cortex or hippocampus at 1 day or 1 month post impact (Supplementary Fig. 4C, D).

These results demonstrate that the acute cognitive deficits of HF-HI are correlated with a change in the synaptic transcriptome, but not with inflammation, p-tau, or $A\beta$.

**Fig. 1 HF-HI impairs cognitive function and alters synaptic transcriptomic signatures. a** Schematic of the experimental design for this figure. Mice were subject to sham or HF-HI before behavioral and RNA-seq analysis at 24 h after the final impact. **b** T-maze cognitive assessment showed a decrease in working memory at 24 h post HF-HI (mean ± SEM; $n = 12$ per group, $*P = 0.048$, unpaired, two-tailed $t$ test). **c** Hierarchical clustering of cortical RNA-seq analysis using Euclidean distance reveal a separation of sham and HF-HI (sham $n = 10$, HF-HI $n = 11$). **d** Principal component analysis (PCA) also shows clear separation of sham and HF-HI brain at 24 h post injury (sham $n = 10$, HF-HI $n = 11$). **e** Volcano plot showing the spread of 3836 differentially expressed genes (1880 upregulated and 1956 downregulated). **f** Directionality dot plot of gene sets obtained by functional enrichment of significantly altered genes in HF-HI mice, using cellular components (CCs) from gene ontology (GO) focusing on synaptic components show that these synaptic families are being modulated, appearing as significant categories in both the up- and downregulated CC. Node size corresponds to the number of genes in each ontology term. Color of each dot is the gene ratio compared to the total gene number. **g** Heatmap displaying the top differentiated genes synaptic signaling biological process (BP) and **h** long-term potentiation Kyoto Encyclopedia of genes and genomes (KEGG) categories show that genes in these categories are being modulated, and not just downregulated. Color scale bar values represent $Z$ value (normalized standard deviations from the mean) for expression relative to the overall mean expression.

**HF-HI induces acute synaptic changes that impair synaptic plasticity.** Based on our transcriptomics and PCR data, we focused on the excitatory synapses as a potential target of HF-HI. We first studied synapse number, but found no change in the dendritic spine density in the cortex or hippocampus of HF-HI mice (layer II/III of the cortex, CA1, and dentate gyrus of the hippocampus) (Fig. 2b and Supplementary Fig. 5A, C). Field recordings from the Schaffer collaterals of acute hippocampal slices from 24 h mice showed that there was no change in synaptic recruitment between groups as the stimulus input/output curves were identical for sham and HF-HI mice (Fig. 2c); however, following high-frequency stimulation, there was a significant blunting of the LTP response in HF-HI brains with a significant head impact and time interaction ($F_{(129, 1806)} = 1.263$; $P = 0.0282$) (Fig. 2d).

To gain insight into the mechanisms that underlie deficient plasticity in HF-HI mice, we investigated glutamatergic synaptic responses. Using paired-pulse ratio recordings of evoked excitatory postsynaptic currents (EPSCs) in the presence of 25 μM bicuculline methobromide (BMR), we found no difference between HF-HI and sham mice, indicating that presynaptic vesicular release is not changed in the HF-HI brain (Fig. 2e). Upon the addition of 1 μM tetrodotoxin (TTX), we observed a significant increase in the miniature EPSCs (mEPSCs) inter-event interval in HF-HI mice (i.e., a decrease in mEPSC frequency), which occurred in the absence of changes to mEPSC amplitude (Fig. 2f–h). To study this further, we recorded the α-amino-3-hydroxy-5-methyl-4-isoxazolepropionic acid (AMPA)-to-N-methyl-D-aspartate (NMDA) ratio in CA1 neurons. We found a decrease in the AMPA/NMDA ratio in HF-HI mice at 1 day (Fig. 2i; $P = 0.023$), suggesting a reduction in surface AMPA receptors vs NMDA receptors following HF-HI.

To determine if HF-HI alters the composition or relative expression of extrasynaptic NMDA receptors, we used a high-frequency stimulation protocol to assess the relative decay of extrasynaptic receptors vs synaptic receptor decay. We found no difference between sham and HF-HI groups (Supplementary Fig. 5D). The percent normalized charge transfer of the extrasynaptic receptor decay also revealed no difference between the two groups (Supplementary Fig. 5E). In addition, we measured the tonic NMDA current, which is mediated by extrasynaptic NMDA receptors and influenced by alterations in the function of glial glutamate transporters[21]. We found no difference between groups (not shown) and infer from these data that HF-HI is not changing glial glutamate transporter capabilities.

We next sought to address whether HF-HI altered the intrinsic properties of CA1 pyramidal neurons to contribute to overall network dysfunction. To study neuron excitability, we used the current clamp with 20 pA steps and observed the firing rate of CA1 neurons from sham and HF-HI mice. We found a significant increase in the approximate rheobase of HF-HI neurons compared to sham neurons (Fig. 2l; $P = 0.0019$). We studied the firing rate of these neurons as we increased the current and found a significant effect of increasing current ($F_{(9,126)} = 77.69$; $P < 0.0001$), a significant effect of head impact ($F_{(1,14)} = 11.24$; $P = 0.0047$) and a significant interaction between the two ($F_{(9,126)} = 7.698$, $P < 0.0001$). Post hoc analysis revealed a significant reduction in neuron excitability in HF-HI mice (Fig. 2j, $P < 0.01$). We used hyperpolarizing current injections to determine changes in the voltage responses that are related to the hyperpolarization-activated current and specific potassium currents. We found increased $V_1$ rectification, which is indicative of inward rectification of potassium currents at 1 day post HF-HI ($P = 0.049$; Fig. 2m), but no change in $V_2$ sag between the two groups (Fig. 2n). We also did not observe any changes to cellular input resistance or capacitance (Supplementary Fig. 5F, G).

Together, these data show that the acute cognitive deficits and synapse transcriptome changes of HF-HI occur alongside a decrease in synaptic function, neuron excitability, and synaptic plasticity.

**HF-HI causes chronic cognitive deficits in mice.** We have previously reported that 30 head impacts delivered with low frequency (30 impacts delivered over 6 weeks) does not produce cognitive dysfunction[14]. To assess the effect of a higher frequency of impacts on chronic cognitive function, we exposed three different cohorts of mice to HF-HI, allowed them to recover for a month, and then tested cognition using a different test for each cohort. In the T-maze, HF-HI mice display impaired working memory through fewer spontaneous alternations compared to sham mice (Fig. 3b, $P = 0.0356$).

In the Barnes maze, mice exposed to HF-HI showed significant deficits during the acquisition phase compared to sham animals (Fig. 3c), with a significant effect of head impact ($F_{(1, 38)} = 9.379$; $P = 0.004$) and time ($F_{(3, 114)} = 81.57$; $P < 0.0001$). Post hoc analysis revealed a significant increase in the latency of HF-HI mice to find the escape hole on days 30 and 31 of testing (Fig. 3c; $P < 0.05$). To test spatial memory, we conducted a probe test 72 h after the final training session. HF-HI mice displayed a reduction in the number of entries into escape hole zone ($P = 0.029$) and a significant interaction between head impact and maze quadrant ($F_{(3, 152)} = 6.268$; $P = 0.0005$) showing that HF-HI mice spent less time in the target quadrant ($P = 0.009$) (Fig. 3d–f). There were no differences between the groups in total distance traveled or mean speed during the probe day (Fig. 3g, h).

In the Morris Water maze, mice exposed to HF-HI showed significant deficits during acquisition phase compared to sham animals (Fig. 3i), with a significant effect of head impact ($F_{(1, 40)} = 5.092$; $P = 0.0296$), and time ($F_{(3, 120)} = 24.01$; $P < 0.0001$). Post hoc analysis revealed a significant increase in the latency of

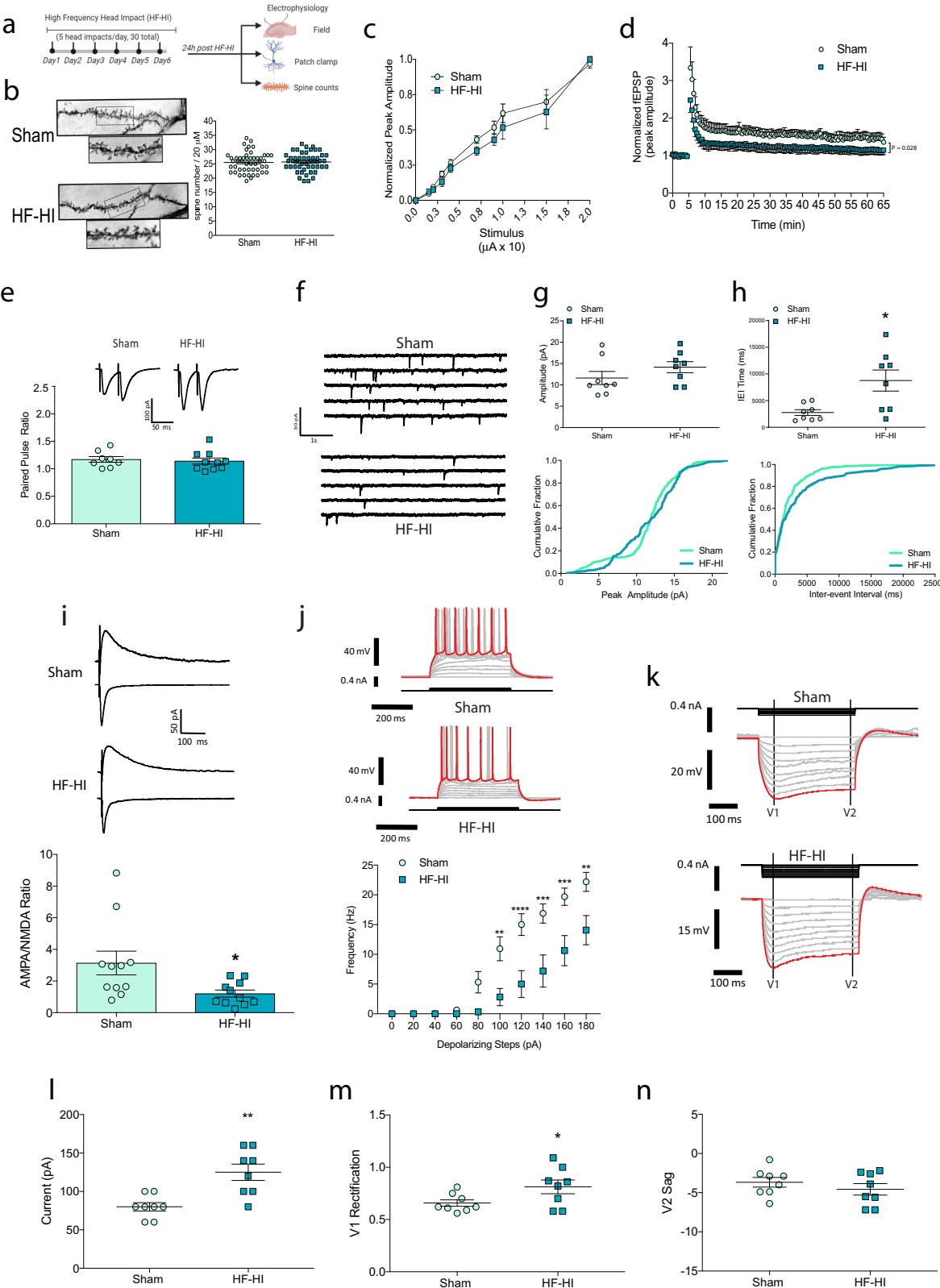

HF-HI mice to find the hidden platform on day 31 of testing (Fig. 3i; $P = 0.016$). To test spatial memory, we conducted a probe test 72 h after the final training session. HF-HI mice displayed a reduction in the number of entries into the platform zone (Fig. 3j, k; $P = 0.006$) and significant interaction between head impact and quadrant on the number of entries ($F_{(3, 160)} =$

2.861; $P = 0.0387$) and time in the target zone ($F_{(3, 160)} = 3.486$, $P = 0.0172$) with post hoc analysis revealing reductions in HF-HI mice compared to sham ($P < 0.05$) (Fig. 3l, m).

Collectively, these data demonstrate that HF-HI induces chronic cognitive deficits in mice. As described earlier, 1 month mice lacked inflammatory phenotypes in all brain areas outside

**Fig. 2 HF-HI reduces synaptic plasticity through modifications to synaptic function. a** Schematic of the experimental design. **b** Dendritic spine analysis of CA1 neurons shows that there was no effect of HF-HI on spine density ($n = 48$ sham and 49 HF-HI neurons from 5 mice/group, insert box = 20 μm). **c** Stimulus input–output curve in the Schaffer collaterals of acute hippocampal slices ($n = 8$) and **d** long-term potentiation (LTP) reveal decreased plasticity in HF-HI hippocampus ($n = 8$ per group). **e** Paired-pulse ratio ($n = 8$ sham, $n = 10$ HF-HI). **f** Example traces of miniature excitatory postsynaptic currents (mEPSC) for sham (top) and HF-HI (bottom) mice. **g** Mean amplitude of mEPSCs per cell and a cumulative fraction of amplitudes were unchanged between groups ($n = 8$ mice per group). **h** Mean inter-event interval of mEPSCs and cumulative fraction of inter-event interval per group showed that mEPSC frequency was decreased in HF-HI CA1 neurons ($n = 8$ per group. *$P = 0.011$, unpaired, two-tailed $t$ test). **i** Example traces of voltage-clamp recordings at −70 or +40 mV to identify AMPA- and NMDA-mediated EPSC. The AMPA/NMDA ratio was decreased in HF-HI CA1 neurons ($n = 11$ per group; *$P = 0.031$, unpaired, two-tailed $t$ test). **j** Representative traces of voltage responses of CA1 neurons to 20 pA step, depolarizing current injections. Frequency of action potential firing in response to depolarizing current injections was decreased in HF-HI CA1 neurons ($n = 8$ per group; **$P = 0.0019$; ***$P = 0.001$, ****$P < 0.0001$, two-way repeated-measures ANOVA with Bonferroni post hoc). **k** Example traces of voltage responses in CA1 neurons to 20 pA step, hyperpolarizing current injections for sham (top) and HF-HI (bottom) mice. **l** Approximate rheobase was increased in HF-HI CA1 neurons ($n = 8$ per group; unpaired, two-tailed $t$ test, **$P = 0.0019$). **m** $V_1$ rectification indicating the inward rectifying potassium currents was increased in HF-HI CA1 neurons ($n = 8$; unpaired, two-tailed $t$ test, *$P = 0.049$), but **n** sag of the voltage response indicating Ih current ($V_2$ sag) was unchanged ($n = 8$). Data are shown as mean ± SEM.

the optic tract and did not have accumulations of $A\beta_{40}$ or p-tau (Supplementary Fig. 3 and 4).

**Transcriptomics reveal the synapse as a primary target of HF-HI.** We repeated our RNA-seq analysis in 1-month HF-HI mice. Hierarchical clustering of the aligned and normalized read counts indicated that our two treatments (sham and HF-HI) clearly separated in terms of overall gene expression (Fig. 4a) with 908 significantly differentiated genes at 1 month (Fig. 4b). We used GO enrichment to gain insights into the molecular mechanisms and pathways that might be contributing to chronic cognitive dysfunction. Analysis of CC categories revealed synapse (GO:0045202), with neuron projection (GO:00430005) and multiple synaptic modalities, including post synapse (GO:0098794) and asymmetrical synapse (GO:0032279) highly represented (Fig. 4c). Analysis of BP categories revealed multiple synaptic signatures, including synaptic signaling (GO:0099536), anterograde synaptic signaling (GO:0098916), and chemical synaptic transmission (GO:0007268) groups (Fig. 4c). Directionality analysis showed that the majority of these pathways were not simply upregulated or downregulated, but instead were modified such that the synaptic pathways and CCs appeared in both the significantly upregulated and significantly downregulated pathways (Supplementary Fig. 6B). Gene analysis revealed a robust up- and downregulation of individual genes involved in the synaptic compartment of neurons (Supplementary Fig. 6C), and to genes involved in learning and memory (Fig. 4d) and synaptic signaling (Fig. 4e).

We compared the differentially expressed genes in the 1-month HF-HI cohort to the 1-day HF-HI cohort, and found that 52.6% of the 1 month genes was also differentially expressed at 24 h (Fig. 4f), with 42.1% of the 1-month synaptic compartment genes also found at 1 day, and 50% of the synaptic signaling genes from 1 month also found at 1 day (Fig. 4f, g). These analyses reveal a subset of synaptic genes, including AMPA subunit and AMPA-trafficking genes, and genes involved in calcium signaling, which are altered both acutely and chronically by HF-HI.

Together, the 1-day and 1-month transcriptome data reveal a strong acute and chronic synaptic phenotype in HF-HI mice that points to synaptic modification rather than synaptic loss.

**Transcriptome analysis reveals an impaired synaptic function in CTE brain.** To evaluate if our synaptic signature in HF-HI mouse has relevance to human pathophysiology, we assessed a human transcriptomic study of six late-stage postmortem CTE brains[22]. We found 134 genes in common between 1-day HF-HI mice, 1-month HF-HI mice, and CTE brains (Fig. 5a) and identified a strong synaptic component shared by all three groups

when studying the overlap of all significantly changed BPs and cellular components (Supplementary Fig. 7A–D). Genome-wide association study (GWAS) analysis of the common genes showed significant overlap in a number of cognitive and affective order diseases including Alzheimer's disease, Parkinson's disease, depression, and post-traumatic stress disorder (Fig. 5b). The original study of these brains used weighted gene co-expression network analysis and identified modules of altered genes that were correlated with CTE, including a neuron-enriched module (identified as the "blue module" in the original manuscript)[22]. Analysis of this cluster using updated GO functional categories revealed that human CTE subjects display significant alterations in multiple synaptic processes and neuronal components, including anterograde trans-synaptic signaling, chemical synaptic signaling, synaptic signaling (BP), and neuron part, synapse, and synapse part (CC) (Fig. 5c). When this blue module was compared to the HF-HI mice, we found that 59% of the common genes identified in Fig. 5a were from this neuronal cluster (Fig. 5d), and KEGG and pathway analysis revealed multiple overlapping targets including glutamatergic synapses and diseases including Alzheimer's disease (Fig. 5e and Supplementary Fig. 8). In contrast, a comparison of genes from the CTE inflammation cluster revealed that there were no shared genes between the CTE brain and the HF-HI mice (Fig. 5f, g).

These data demonstrate that CTE brains and HF-HI mice have strong synaptic phenotypes, but do not share an inflammatory phenotype.

**HF-HI induces chronic synaptic changes that impair synaptic plasticity.** To assess which of the electrophysiological phenotypes remained alongside the chronic cognitive and transcriptome changes, we repeated our 1-day field and patch studies in 1-month HF-HI mice. Field recordings from the Schaffer collaterals of acute hippocampal slices from 1 month HF-HI mice showed that there was no change in synaptic recruitment as the stimulus input/output curves were identical for sham and HF-HI mice (Fig. 6b); however, following high-frequency stimulation, there was a significant blunting of the LTP in HF-HI brains with a significant head impact and time interaction ($F_{(128, 2048)} = 1.329$; $P = 0.0097$) (Fig. 6c).

Similar to the 24 h mice, we found no effect of HF-HI on paired-pulse ratio at 1 month, indicating that presynaptic vesicular release remains unaffected in the chronic HF-HI brain (Fig. 6d). We again found no change in mEPSC amplitude, and at the 1 month time point, the mEPSC inter-event interval had returned to sham levels (Fig. 6e–g). The only synaptic phenotype that was significantly different at both the 24 h and 1 month time point was the AMPA/NMDA ratio in HF-HI mice (Fig. 6h; $P =$

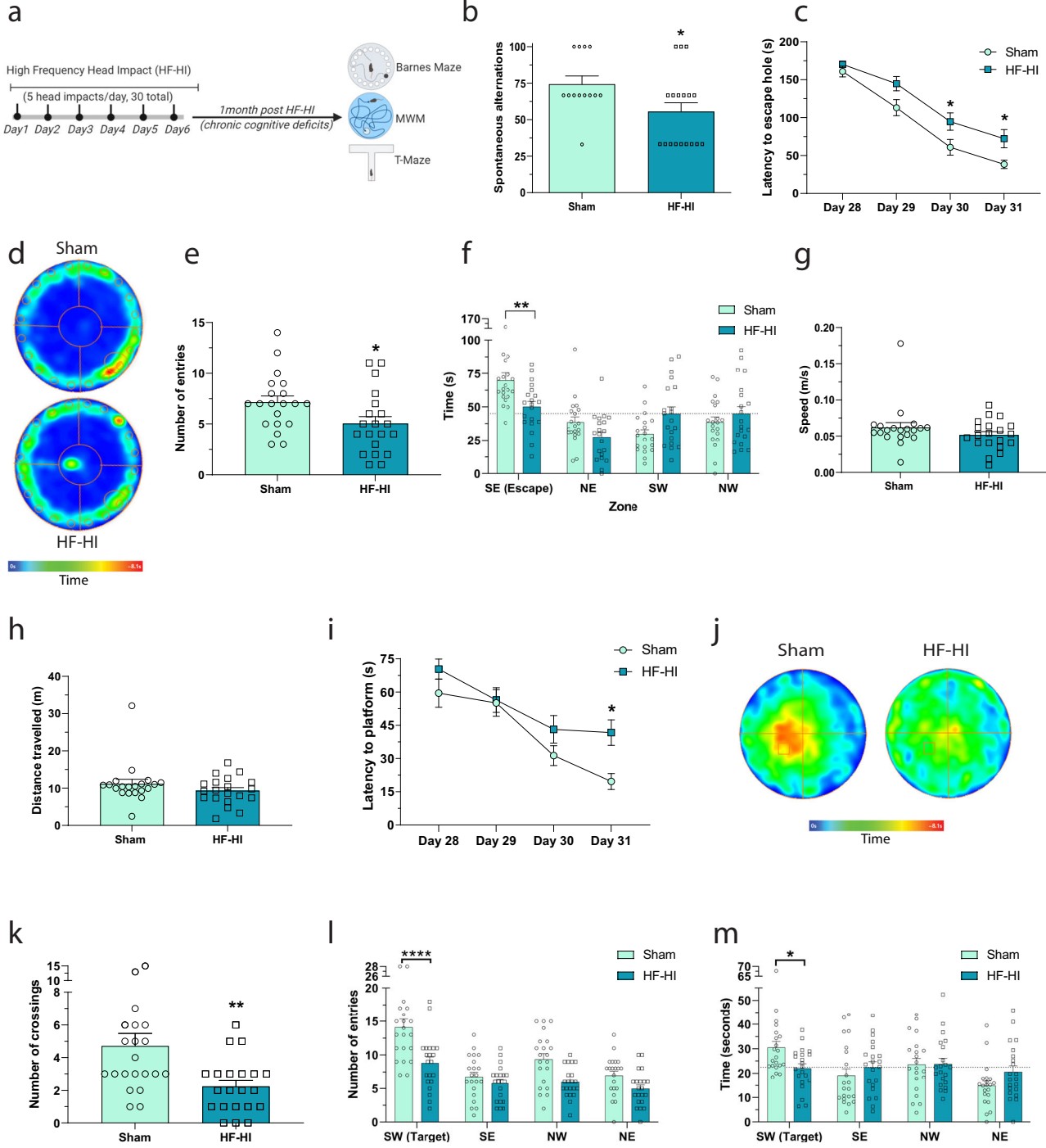

**Fig. 3 HF-HI impairs chronic cognitive function. a** Schematic of the experimental design for this figure. Mice were subject to sham or HF-HI before cognitive assessment at 1 month after the final impact. **b** T-maze cognitive assessment at 1 month showed decreased working memory in HF-HI mice ($n = 13$ sham, $n = 18$ HF-HI, *$P = 0.0356$, unpaired, two-tailed $t$ test). **c** Latency to platform during the acquisition phase of the Barnes maze shows decreased learning in HF-HI mice. Data from two trials per day on 4 consecutive days ($n = 20$ per group, *$P = 0.04$, two-way RM ANOVA with Bonferroni post hoc test). **d** Heatmaps detailing group time spent in Barnes maze during the probe trial at 72 h post acquisition. **e** Number of entries into escape target area during the 72 h delay probe trial is reduced in HF-HI mice ($n = 20$ per group, *$P = 0.029$, unpaired, two-tailed $t$ test). **f** Time spent in Barnes maze escape quadrant is reduced in HF-HI mice during the 72 h delay probe trial ($n = 20$/group, **$P = 0.009$, two-way ANOVA with Bonferroni post hoc test). **g** Speed and **h** distance traveled is similar between groups ($n = 20$ per group). **i** Latency to platform is increased in HF-HI mice during the acquisition phase of the Morris Water maze (MWM). Data from two trials per day on 4 consecutive days ($n = 21$/group, *$P = 0.016$, two-way RM ANOVA with Bonferroni post hoc test). **j** Heatmaps detailing group time spent in MWM during the probe trial. **k** Number of platform crossings is reduced in HF-HI mice during the 72 h delay probe trial ($n = 21$ per group, **$P = 0.006$, unpaired, two-tailed $t$ test). **l** Number of entries into the target quadrant was reduced in HF-Hi mice during the 72 h delay probe trial ($n = 21$ per group, ****$P < 0.0001$, two-way ANOVA with Bonferroni post hoc test). **m** Time spent in MWM quadrants showed decreased time in the target zone for HF-HI mice during 72 h delay post acquisition ($n = 21$ per group, *$P = 0.048$, two-way ANOVA with Bonferroni post hoc test). Data are shown as mean ± SEM.

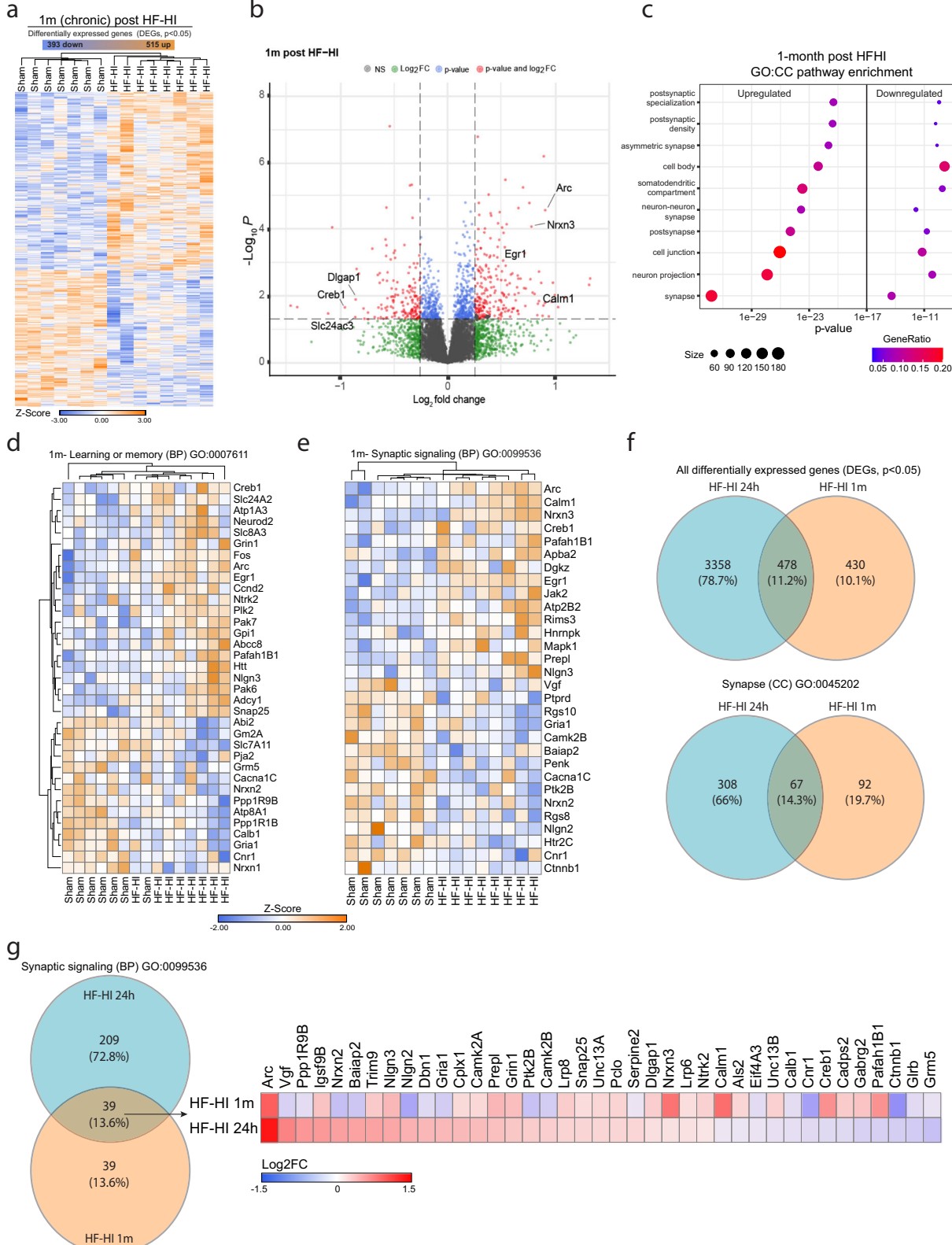

0.019), demonstrating an imbalance in the excitatory receptors in HF-HI CA1 neurons.

We next sought to examine if the decreased excitability of CA1 pyramidal neurons could be contributing to memory loss in chronic HF-HI mice. In contrast to the 24 h mice, we found no effect of HF-HI on neuron excitability at 1 month (Fig. 6i), with approximate rheobase and $V_1$ rectification returning to normal

(Fig. 6k, l). We also did not observe any changes in $V_2$ sag (Fig. 6m), cellular input resistance (Supplementary Fig. 5H), or capacitance (Supplementary Fig. 5I).

Together, these data show that the only chronic electrophysiology deficits that remain at 1 month post HF-HI alongside chronic cognitive deficits are the decreased plasticity and a reduction in the AMPA/NMDA ratio. Earlier phenotypes of

**Fig. 4 Cortical transcriptomic signature at 1 month after HF-HI.** RNA-seq analysis 1 month after HF-HI. **a** Heatmap with all significantly differentially expressed genes (DEG) (515 upregulated and 393 downregulated) of sham and HF-HI cortical genes identified by RNA-seq show clustering by group. Color scale bar values represent $Z$ value (normalized standard deviations from the mean) for expression relative to the overall mean expression ($n =$ 7 sham, 8 HF-HI). **b** Volcano plots show the spread of these significantly differentially expressed genes. **c** Directionality dot plot of gene sets obtained by functional enrichment of significantly altered genes in HF-HI mice, using cellular components (CC) from gene ontology (GO) focusing on synaptic components show that these synaptic families are being modulated, appearing as significant categories in both the up- and downregulated CC. Node size corresponds to the number of genes in each ontology term. Color of each dot is the gene ratio compared to the total gene number. **d, e** Heatmaps displaying the top differentiated genes in learning or memory and synaptic signaling biological process (BP) categories show that genes in these categories are being modulated by HF-HI, and not just downregulated. Color scale bar values represent $Z$ value (normalized standard deviations from the mean) for expression relative to the overall mean expression. Individual samples are listed below each heatmap. **f** Venn diagram demonstrating overlap of all significantly DEGs at 24 h and 1 month after HF-HI. Venn diagram demonstrating the overlap of synapse CC genes at 24 h and 1 month post HF-HI. **g** Venn diagram demonstrating the overlap of synaptic signaling BP genes between the 24 h and 1 month time points with a heatmap demonstrating the Log 2 fold change and directionality of the 39 conserved genes.

reduced mEPSC frequency, abnormal potassium channel rectification, and impaired neuron excitability had all spontaneously resolved by 1 month post impact.

**HF-HI causes synaptic modification and cognitive deficits through a glutamate and NMDA receptor-dependent mechanism.** Glutamate is increased after head impact in both human athletes and animal models[23–25]. The specific changes to the ratio of glutamate receptors is strongly suggestive of a glutamate-centric event causing synaptic modification. To test the hypothesis that HF-HI causes synaptic modifications in response to repetitive temporary elevations in glutamate, we pretreated mice each day with memantine, an extrasynaptic NMDA receptor antagonist, 1 h prior to head impacts. This was a "proof-of-concept" experiment to demonstrate that the acute synaptic changes of HF-HI were being driven by glutamate and extrasynaptic NMDA receptors, and not by other factors such as inflammation.

Prophylactic memantine treatment blocked the development of working memory impairments in the T-maze that was seen in saline-treated HF-HI mice (Fig. 7b). Analysis of variance (ANOVA) revealed a significant effect of head impact ($F_{(1, 38)} =$ 4.674; $P = 0.037$), with post hoc analysis demonstrating that HF-HI saline-treated HF-HI mice had significantly lower spontaneous alternation ($P = 0.029$), but memantine-treated HF-HI mice had normal spontaneous alternation (Fig. 7b).

Our earlier work revealed that a reduced AMPA/NMDA ratio is a critical functional synaptic consequence of HF-HI, and the only functional assay that is altered at both 1 day and 1 month post impact. We tested if memantine pretreatment could prevent the development of this receptor imbalance in CA1 pyramidal cells. ANOVA revealed a significant difference between groups ($F_{(2, 24)} = 8.198$, $P = 0.002$) with post hoc analysis showing a significantly reduced AMPA/NMDA ratio in vehicle HF-HI mice ($P = 0.016$) that were completely prevented in memantine HF-HI mice ($P = 0.003$) (Fig. 7c, d).

Memantine pretreatment also prevented many of the transcriptomic changes involved in the synaptic signature of HF-HI (Fig. 7e, f). Analysis revealed that 674 genes were altered in saline HF-HI mice compared to saline sham mice, while 338 genes were altered in memantine HF-HI mice compared to memantine sham mice, with an overlap of only 7.2% of the total genes (Fig. 7g). We found that memantine significantly attenuated the effect of HF-HI on the top 20 most upregulated and downregulated genes from the saline HF-HI mice (Fig. 7g). GO pathway enrichment for CCs and BPs revealed that HF-HI once again significantly affected synaptic compartments, but we found that memantine significantly blunted the impact of HF-HI on these pathways. Memantine significantly reduced both the number of genes and the gene ratio in each synaptic category compared to saline HF-HI mice (Fig. 7h and Supplementary Fig. 6d). The reversal effect

of memantine on the individual genes important to glutamatergic synapses, learning and memory, and synaptic signaling was confirmed (Fig. 7i and Supplementary Fig. 6e, f).

To determine if the acute effects of memantine on the transcriptome and synaptic function translated to chronic changes in behavior, we pretreated mice with memantine and probed their chronic cognitive function using the T-maze and Barnes maze at 1 month post impact. Memantine pretreatment prevented the development of chronic working memory impairments in the T-maze that were observed in saline-treated HF-HI mice (Fig. 8b). ANOVA revealed a significant effect of impact ($F_{(1, 54)} = 5.555$; $P = 0.022$), with post hoc analysis demonstrating that saline-treated HF-HI mice had significantly lower spontaneous alternation ($P = 0.013$), but memantine HF-HI mice did not (Fig. 8b). In the Barnes maze, ANOVA revealed an effect of time ($F_{(3,165)} = 150.1$; $P < 0.001$) and post hoc analysis revealed a significant difference between saline HF-HI and memantine HF-HI mice, but only on day 30 of training ($P = 0.012$) (Fig. 8c). The probe trial was conducted 72 h later, and ANOVA revealed a significant interaction of drug and impact on the time in the target zone ($F_{(1,54)} = 8.084$; $P = 0.063$) with saline HF-HI spending significantly less time in the target zone ($P = 0.014$) and memantine significantly reversing this effect ($P = 0.015$) (Fig. 8d–f). ANOVA also revealed a significant interaction of impact group with maze quadrant ($F_{(9,216)} = 4.509$, $P < 0.0001$) with saline HF-HI spending significantly less time in the target quadrant ($P = 0.001$) and memantine significantly reversing this effect ($P = 0.039$) (Fig. 8g). There was no effect of HF-HI or memantine on distance traveled or mean speed (Fig. 8h, i).

These data show that memantine pretreatment blocks the development of the key synaptic transcriptome adaptations, synaptic function, and acute working memory deficits of HF-HI.

## Discussion
Our study identifies synaptic adaptation in response to HF-HI as a key determinant of chronic cognitive dysfunction in mice. We have developed a mouse model that allows us to differentiate the physiological events that occur in response to head impact separately from more severe pathological responses of TBI. We carried out a comprehensive analysis of these mice and determined that adaptation of the synapse in response to head impact is sufficient to trigger cognitive dysfunction, and this altered brain state remains long after the impacts have been discontinued. The transcriptome signatures we identified as being modified belong to the same neuronal compartments that are degraded in the CTE brain, validating the translational importance of our research. These data are consistent with synaptic adaptation causing strong cognitive deficits following head impact, and identify synaptic dysfunction as a target for cognitive impairments caused by repeated head impacts.

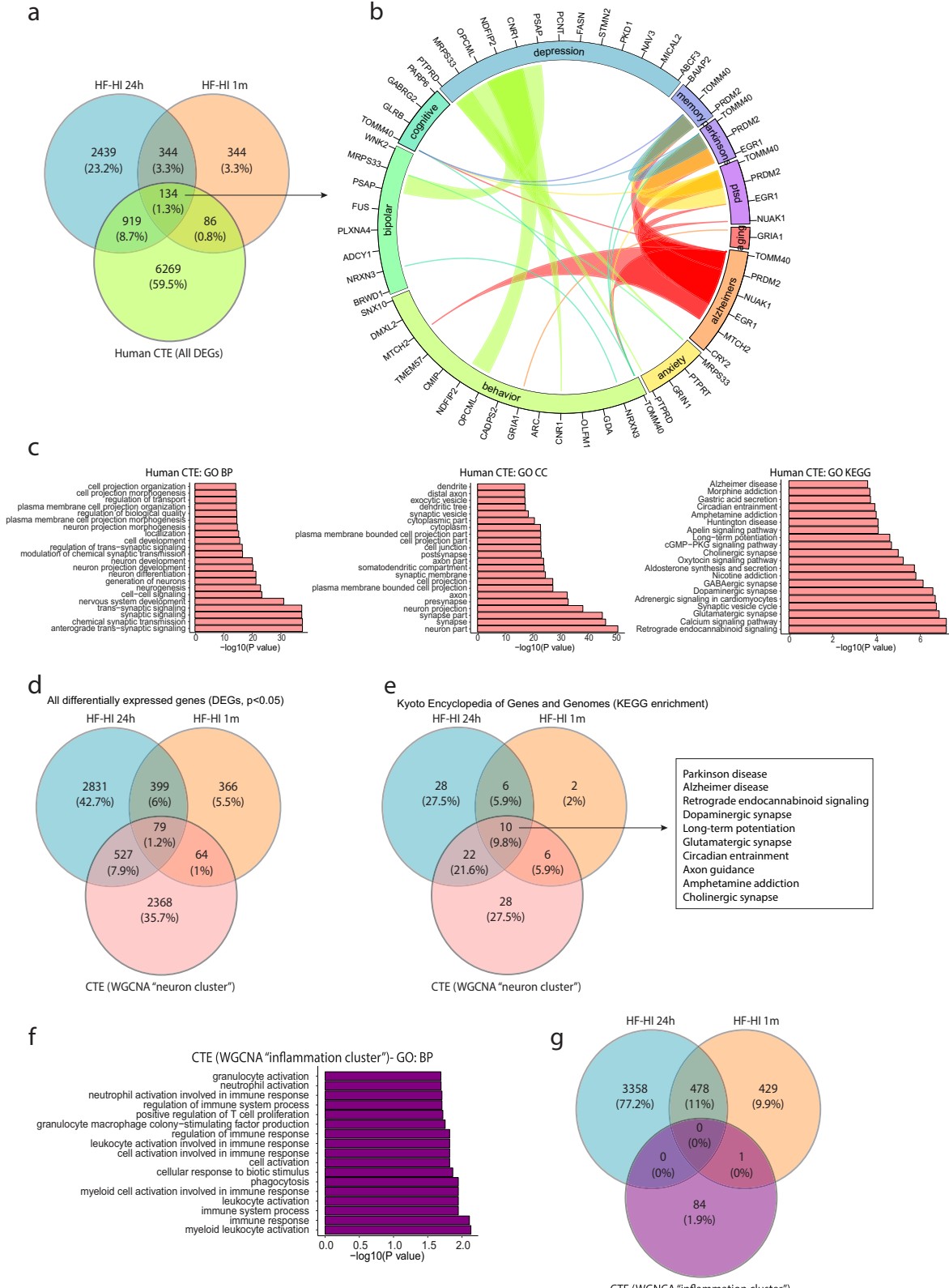

Behavioral dysfunction in rodent models of TBI is well documented. Severe models of brain injury, such as controlled cortical impact, present with widespread axonal injury, cell death, neuroinflammation, and behavioral deficits that increase with the injury severity[26]. Head impact paradigms more analogous to the human condition with respect to maintenance of structural integrity and injury conditions are closed-skull models[27]. Many of these models present no overt tissue damage, cell death, or lesion formation but do present with white matter degeneration, neuroinflammation, long-term learning and memory deficits, and increased anxiety- and depressive-like behavior[6,9,13,16,28–33]. Of these studies, those that have measured p-tau levels have reported no changes despite the manifestation of symptoms[34–36]. The majority of these models do not exceed five head impacts total,

**Fig. 5 Altered synaptic signatures are prominent features in postmortem CTE brains and HF-HI mice. a** Venn diagram of significant differentially expressed genes from HF-HI brains and all weighted gene correlation network analysis (WGCNA) clusters from postmortem human CTE brains identifies 134 overlapping genes. **b** Circos plot depicting overlapping differentially expressed genes in human CTE brains and HF-HI mice (FDR < 0.1) and how they associated with neurodegenerative diseases or cognitive traits as identified in GWAS studies. The proportion of the circle's circumference allocated to each disease represents the number of genes associated with that disease that are also differentially expressed in both human CTE brain and HF-HI mice. The lines connecting genes within the circle indicate which genes were shared amongst disease and trait signatures. **c** Gene sets obtained by functional enrichment of WGCNA–neuron cluster from postmortem human CTE brains using gene ontology (GO) to display significant biological processes (BP), cellular components (CCs), and Kyoto encyclopedia of genes and genomes (KEGG) functional enrichment show changes to synaptic compartments. **d** Venn diagram demonstrating overlap of differentially expressed genes in human CTE WGNCA–neuron cluster and mouse HF-HI brains. **e** Venn diagram demonstrating overlapping KEGG pathways between human CTE WGNCA–neuron cluster and HF-HI brains. The ten overlapping pathways are listed. **f** Inflammatory pathways analyzed from the postmortem CTE WGCNA–Inflammation cluster using gene ontology BP analysis, and **g** Venn diagram comparison of inflammatory genes compared to HF-HI mice show no overlap between CTE and HF-HI mice.

and many have separated the impacts with an interval of 24–48 h, making the identification of any influence of a high frequency of impacts on altered function difficult to infer. Our model attempts to address the role of frequency of head impacts in deficit development and allows us to illustrate a mechanism unique to HF-HIs that may underlie learning and memory impairment following exposure. The lack of structural damage, hyperphosphorylated tau, and Aβ accumulation in our model further helps us explore the idea that a high number of hits may engage physiological changes that contribute to the cognitive symptoms associated with concussive head injuries. We can directly assess the effect of HF-HI by comparing our results to our previous work on 30 head impacts delivered with lower frequency[14]. This lower frequency model presented with similar optic tract pathology to the HF-HI model, but did not display cognitive impairments. We, therefore, used the unique combination of the presence of a strong cognitive phenotype and a lack of TBI pathology to ask how behavioral events might arise absent neuropathological events.

We used RNA-seq to provide an unbiased transcriptome analysis of the HF-HI model. With multiple potential mechanisms of action possible in HF-HI mice, we chose to work with whole homogenate instead of single-cell sequencing to ensure we could fully assess the role of cell death and neuroinflammation pathways. GO analysis revealed that synapses are a major target of HF-HI, with synaptic transmission being the top BPs affected, and glutamatergic synapses identified as a significantly impacted KEGG pathway. Directionality analysis revealed that multiple components of the synaptic compartments were both up- and downregulated, demonstrating that a synaptic adaptation—and not just a downregulation of synapses as would be expected in a more severe injury model. A recent re-analysis of CTE sequencing data also identified synapses as an affected compartment in disease[22,37], and when we compared our synaptic signature to the synaptic signature from late-stage CTE brains, we found that the same synaptic compartments that adapt in the HF-HI brain are similar to those that are degraded in CTE.

The transcriptome signature from HF-HI mice has demonstrated that it is a very different model compared to existing TBI and mTBI models. We did not find a significant signature for inflammation or cell death pathways in our mice. In contrast, transcriptome analysis of a single severe contusion injury in mice or a single mild lateral fluid percussion (LFP) injury in rats shows that inflammation is heavily involved in the post-injury response[38–40]. Areas of overlap do exist between HF-HI mice and more severe injury models, with signatures associated with Alzheimer's disease, depressive disorder, and intelligence[39], and pathways involved in LTP and synaptic transmission being similarly affected[41]. Intriguingly, single-cell analysis from the hippocampal cells after LFP injury reveals a similar signature compared to our whole homogenate analysis[40]. We conclude that

the synaptic changes in HF-HI mice are a key overlapping molecular event that is also observed in many different animal models of TBI and in the CTE brain, and we show that changes to these compartments can occur in the absence of activation of inflammation or cell death signatures.

A loss of hippocampal plasticity is a common event in TBI models of all severities[11,42–46]; however, the combination of pathologies that occur in TBI include neuron cell death, loss of axon integrity, reduced synapse number, and synaptic adaptation can all affect plasticity. Determining the exact role of individual pathologies in TBI-induced plasticity alterations is challenging. An in-depth study into the loss of LTP in the CA1 of LFP mice demonstrated that TBI completely inhibited the induction of LTP, and this was due to a reduction in the NMDA potentials[47]. This loss of LTP was accompanied by a shift in hippocampal excitability where the CA1 region had a decrease in the input/output curve, and an increase in the inhibitory postsynaptic current[48]. LFP is a more severe TBI model than our HF-HI model, and the decrease in LTP that occurs in HF-HI mice happens without changes to the input/output curve. This same phenotype (reduced LTP, no change to input/output curve) has also been reported in a single impact mTBI model[42], suggesting that more discrete mechanisms are involved in the reduction in plasticity in mTBI models compared to severe TBI models.

To determine the role of NMDA and AMPA currents in the impaired LTP seen in HF-HI mice, we focused on the AMPA/NMDA ratio. The ratio of AMPA/NMDA is a surrogate that is used to bypass the variability that occurs in direct NMDA and AMPA synaptic-current measurements caused by the positioning of electrodes or the recruitment of synapses by each stimulus pulse[49]. The sustained decrease in the AMPA/NMDA ratio in HF-HI mice indicates that there is a shift in the relative number of AMPA receptors to NMDA receptors, and thus a decrease in the strength of excitatory synapses following head impact. There are a number of mechanisms that might drive this effect including an increase in the immediate early gene, *Arc*, which is involved in AMPA internalization from the synapse surface following NMDA receptor activation[50–52], and we find a sustained increase in *Arc* mRNA at 24 h and 1 month post HF-HI. A second potential mechanism is through activation of NR2B containing NMDA receptors, such as those found at perisynaptic and extrasynaptic sites, which have the ability to reduce synaptic AMPA receptor expression[53], with activation of extrasynaptic receptors via ambient glutamate also able to inhibit neuronal plasticity[54].

Few studies have examined specific glutamatergic receptor properties after injury at the single-cell level. In those that do, the AMPA/NMDA ratio is the opposite of our findings, with the AMPA/NMDA ratio increased following repeated LFP injury[11]. This appears to primarily be due to a decrease in the NMDA receptor-mediated current[55]. Studies have reported a decrease in

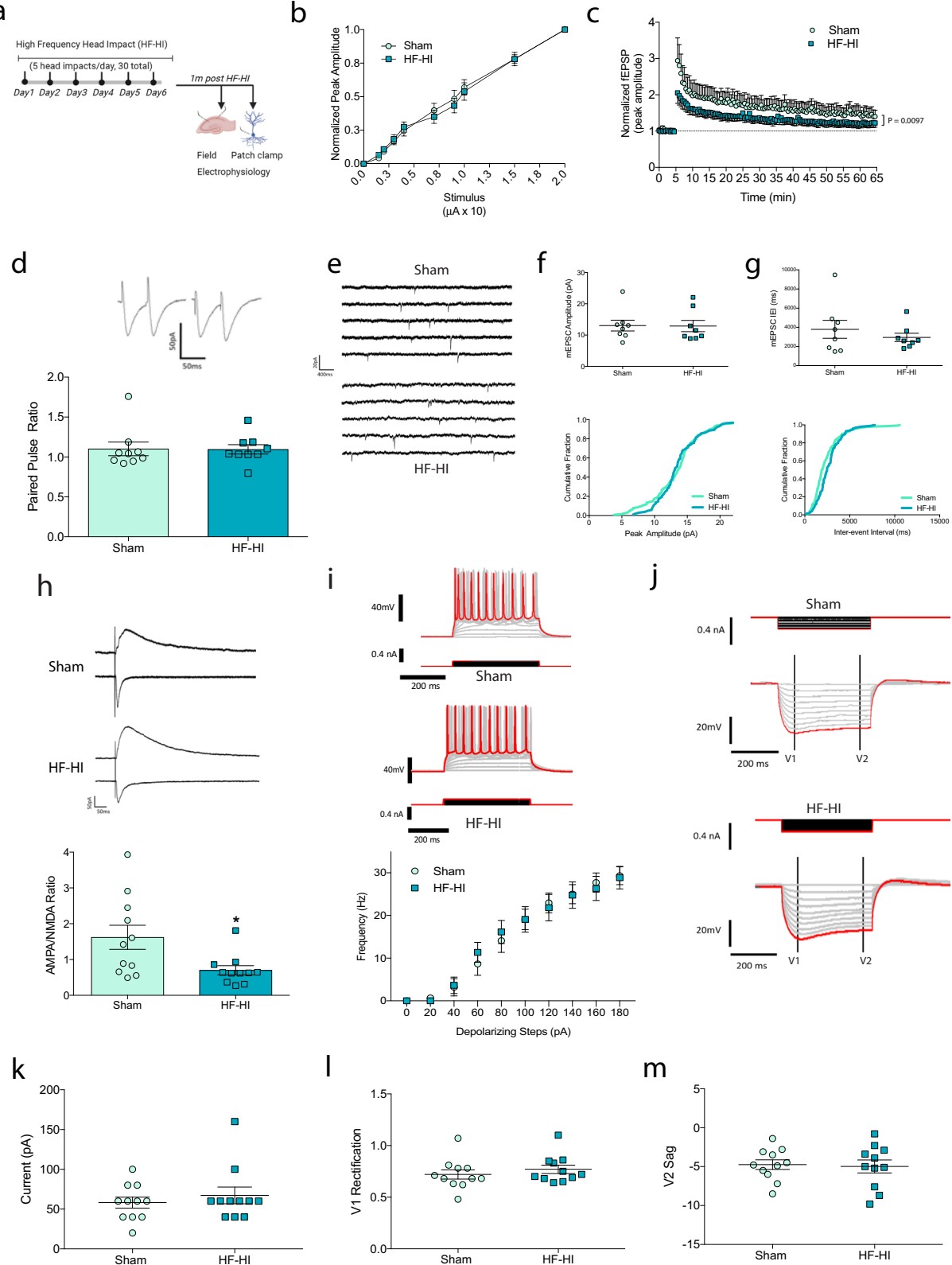

both NMDA- and AMPA-mediated currents[47], and it is possible that this scenario is occurring in our mice with the reduction in AMPA being proportionally more than the reduction in the NMDA current. It should be noted that all the models used in the referenced studies come with excitotoxic neuronal cell loss and axonal injury, which would explain the reduction in NMDA receptors as a compensatory protective mechanism.

Hippocampal neurons are very susceptible to TBI, and neuronal excitability in CA1 neurons has previously been studied in both mild and moderate severe models of TBI. In contrast to our work, most groups report an increase in neuronal excitability following injury[56–62], which we believe is related to the severity of the injury models used. In the controlled cortical impact model of severe TBI, we have previously shown that dendritic spine loss

**Fig. 6 HF-HI changes discrete chronic neuronal and synaptic properties including the AMPA/NMDA ratio and LTP at 1 month post impact. a** Schematic of the experimental design for this figure. **b** Stimulus input–output curve in the Schaffer collaterals of acute hippocampal slices showed no difference in synaptic recruitment between groups ($n = 9$), but **c** long-term potentiation (LTP) reveal decreased plasticity in HF-HI hippocampus ($n = 9$ per group). **d** Paired-pulse ratio estimated from the EPSC recording was elicited with stimuli delivered 50 ms apart ($n = 9$ per group). **e** Example traces of miniature excitatory postsynaptic currents (mEPSC) for sham and HF-HI mice. **f** Mean amplitude of mEPSCs per cell and cumulative fraction of amplitudes did not show any difference between groups ($n = 8$ mice per group). **g** Mean inter-event interval of mEPSCs and cumulative fraction of inter-event interval did not show differences between groups ($n = 8$ mice per group). **h** Example traces of voltage-clamp recordings at $-70$ or $+40$ mV to identify AMPA- and NMDA-mediated EPSC. The AMPA/NMDA ratio was decreased in HF-HI CA1 neurons ($n = 11$ per group; *$P = 0.018$, unpaired, two-tailed $t$ test). **i** Representative traces of voltage responses of CA1 neurons to 20 pA step depolarizing current injections. There was no difference between groups in the frequency of action potential firing in response to depolarizing current injections ($n = 11$ mice per group). **j** Example traces of voltage responses in CA1 neurons to 20 pA step hyperpolarizing current injections for sham (top) and HF-HI (bottom) mice. **k** Rheobase, **l** $V_1$ rectification indicating inward rectifying potassium currents, and **m** $V_2$ sag of the voltage response indicating Ih current, all showed no difference between sham and HF-HI CA1 neurons ($n = 11$ per group). Data are shown as mean ± SEM.

occurs rapidly after injury[63]. Other groups have examined the intrinsic properties of these CA1 neurons in the rat, and show that there is decreased excitability at 1 day post injury[62], but increased excitability in the chronic phase[58]. Both groups focus on the sodium/potassium (Ih) currents as driving the changes in excitability[58,62], demonstrating that the intrinsic properties of severe brain injury neurons fluctuate over time, but remain altered from sham neurons. Temporary decreases in layer V neuron excitability have also been reported in a central fluid percussion mouse model, but this decrease is only seen in axotomized neurons as compared to intact neurons which have increased excitability[61]. In LFP rats, there is an increase in excitability in dentate gyrus neurons that is lost over the course of a month; however, there remains a persistent decrease in threshold to induction of seizure-like electrical activity in the hippocampus after LFP[60]. In our study, we found that CA1 neuron excitability was decreased in HF-HI mice, which occurred alongside an increase in the approximate rheobase, and an increase in inwardly rectifying K$^+$ channels. While this is different from existing reports in more severe TBI models, our data suggest that there is an acute change to K$^+$ channels in the brain of HF-HI mice that occurs in response to mild head impact. Exposure to concussive injuries is associated with increases in extracellular potassium (K$^+$) ion concentration, which can depolarize neurons and perpetuate ionic and neurotransmitter imbalances[64]. Elevated extracellular K$^+$ can increase the conductance of inwardly rectifying K$^+$ channels[65]. The activity of these channels is directly related to the intrinsic excitability of neurons[66–68], Sustaining concussive head impacts within 24 h of one another can delay the recovery of cerebral metabolism[69]; however, several studies on concussive injuries have shown that increased K$^+$ and subsequent metabolic dysfunction resulting from the activation of ion pumps to clear elevated extracellular K$^+$ is very short-lived, and tends to resolve spontaneously after impact cessation[23,64]. In HF-HI mice, the inward rectifying K$^+$ channels normalize and neuronal excitability returns to sham levels by 1 month post impact, supporting the theory that this abnormality is an acute event driven by a transient ionic imbalance caused by head impact.

Release of the excitatory neurotransmitter glutamate is a well-established event in severe models of TBI, with elevations in glutamate lasting for >60 min[70]. Mild concussive blows in rats also lead to the widespread release of glutamate, but it is a short-lived event of a lower magnitude that resolves within minutes of the impact[23,25]. High levels of extrasynaptic glutamate activate cell death pathways through extrasynaptic NMDA receptors[71], but it is unknown how these receptors react to short-lived bursts of glutamate released by minor head impacts. As cell death does not occur in our model, we hypothesize that while glutamate is being released at a quantity where spillover can occur and activate

perisynaptic and extrasynaptic receptors, the amount of glutamate release in HF-HI is not high enough or sustained enough to produce a neurotoxic environment. This is supported by our data showing that HF-HI does not impact the decay of extrasynaptic receptors or the percent normalized charge transfer of the extrasynaptic receptors, which would occur in an excitotoxic phenotype.

A precedent for glutamate-induced synaptic modifications in HF-HI may be found in preconditioning, where low-intensity stressors are used to convey downstream neuroprotection against a subsequent high-intensity stress. Preconditioning is an accepted concept in the fields of ischemia, hypoxia, and epilepsy[72,73], and the mechanism of action of preconditioning varies from acute synaptic adaptation to changes in metabolism and the induction of a hibernation-like state[72–74]. Preconditioning for TBI has been previously hypothesized[75], but all existing studies have used nonimpact preconditioning stimuli[76–79]. Mice preconditioned with a sublethal dose of NMDA are protected against hippocampal cell death and spatial learning deficits after TBI[80]. While we do not know if the synaptic adaptations in HF-HI mice are conveying neuroprotection, the transcriptome signature has similarities to existing preconditioning models. These changes may underlie the concept of acquired resilience in contact sports athletes, whose ability to absorb impacts to the head is thought to improve with exposure conditioning.

To demonstrate the mechanism by which synaptic adaptations occur in HF-HI mice, we conducted a proof-of-concept experiment to block the ability of glutamate to activate extrasynaptic NMDA receptors using the noncompetitive NMDA receptor antagonist, memantine. Memantine's inhibitory profile depends on the likelihood of an NMDA receptor reaching a desensitized state[81,82], which we envision is occurring in peri/extrasynaptic NMDA receptors exposed to elevated glutamate following head impact.

Memantine's standard mechanism of action is as a neuroprotectant. In more severe animal models of TBI and mTBI, memantine spares hippocampal neurons after a single moderate/severe TBI[83], and normalizes NR2B, LTP, APP, p-tau, and neuroinflammation abnormalities in a repeat weight drop model of TBI[46]. In our study, memantine is instead preventing the physiological shift in the synaptic transcriptome and in synaptic function. This results in normalization of the acute synaptic mRNA signature, and prevents the shift in the AMPA/NMDA ratio after HF-HI. More importantly, it prevents the downstream loss of cognitive function in HF-HI mice.

Based on our data, we propose the following synaptic adaptation model to explain cognitive impairments after HF-HI: head impacts cause the release of glutamate at the synaptic cleft, which results in the activation of synaptic and extrasynaptic NMDA receptors, transcriptome modification, and downstream adaptation of the postsynaptic density. Subsets of these transcriptomic and

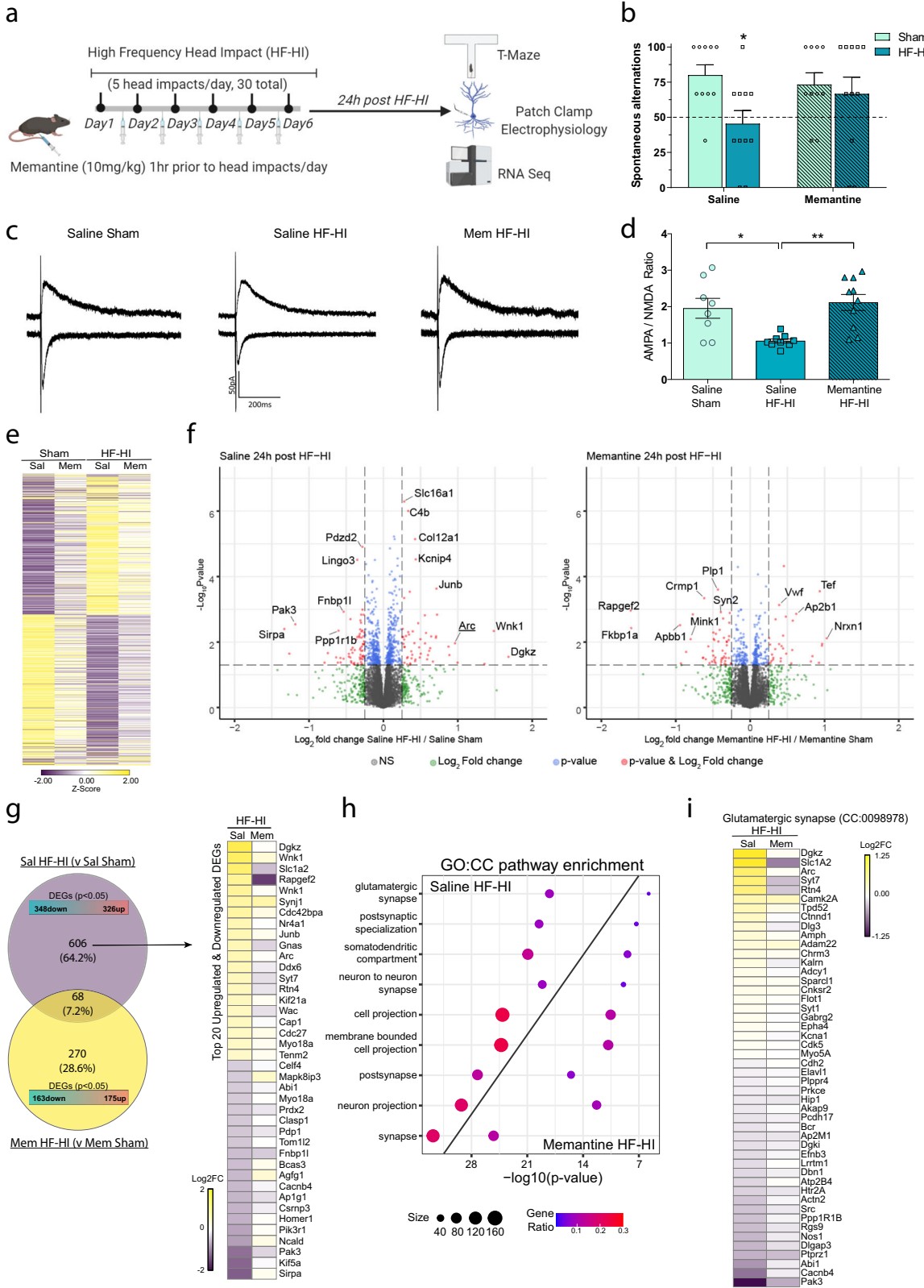

electrophysiological changes are maintained chronically, resulting in reduced synaptic plasticity and impaired learning and memory. We postulate that the synaptic remodeling that occurs in HF-HI mice is a preconditioning event designed to prevent glutamatergic over-stimulation and neuronal cell death; however, these changes also lead to the development of chronic behavioral impairments. To our knowledge, this is the first work that shows that synaptic adaptation

alone is sufficient to cause behavioral changes after head impacts in the absence of structural damage.

Our work uses multiple techniques to demonstrate that behavioral abnormalities after HF-HI are associated with physiological changes to brain synapses that occur independently of synapse loss or p-tau accumulation. Importantly, this does not preclude the formation of CTE pathology from occurring as an

**Fig. 7 The acute cognitive deficits, synapse transcriptome signature, and synapse functional signature of HF-HI are prevented by memantine. a** Schematic of the experimental design for this figure. Mice were pretreated each day with memantine 1 h before sham or HF-HI. Cognitive function, key electrophysiology, and transcriptomic analysis were performed 24 h after the final impact. **b** T-maze revealed a significant effect of HF-HI on cognitive function (*$P = 0.029$) that did not occur in memantine-treated mice (mean ± SEM; sham $n = 10$ per treatment group, HF-HI $n = 11$ per treatment group, two-way ANOVA and Bonferroni post hoc). **c** Representative traces of voltage-clamp recordings at −70 or +40 mV to identify AMPA- and NMDA-mediated EPSC. **d** The AMPA/NMDA ratio was decreased in saline HF-HI CA1 neurons, and reversed with memantine pretreatment (mean ± SEM; $n = 8$ saline sham; $n = 9$ saline HF-HI, $n = 10$ memantine HF-HI; *$P = 0.016$, **$P = 0.0025$, one-way ANOVA with Bonferroni post hoc test). **e** Multivariable four-way group design analysis was performed on RNA-seq data to identify differential gene expression after comparing all interactions and experimental variables of injury (sham/HF-HI) and treatment (saline/memantine). Heatmap showing levels of all significantly altered genes. **f** Enhanced volcano plots constructed for saline HF-HI/saline sham and memantine HF-HI/memantine sham groups display the spread of significant differentially expressed genes (DEG) from each group. **g** Venn diagram demonstrating DEGs identified by RNA-seq. Heatmap demonstrating the effect of memantine on the top 20 up- and downregulated genes from the saline sham vs saline HF-HI group. Scale bar represents the Log 2 fold change compared to the respective sham treatment group. **h** Directionality dot plots of gene sets obtained by functional enrichment significantly altered cellular components (CCs) show that memantine blunts the effect of HF-HI on synapse-related compartments by reducing the number of significantly altered genes, and reducing the gene ratio when compared to saline HF-HI mice. **i** Heatmap displaying the top differentiated genes in glutamatergic synapse CC categories. Each gene is colored by group $Z$-score. Individual genes are listed vertically.

independently formed pathology. The CCs where our synaptic modifications occur are the same synaptic compartments that are lost in postmortem CTE brain. Such transcriptionally modified synapses provide a future target for the treatment of behavioral abnormalities in athletes exposed to HF-HI.

## Methods

**HF-HI model.** All procedures were performed in accordance with protocols approved by the Georgetown University Animal Care and Use Committee. Six- to twelve-week-old male C57Bl/6J mice were housed 5 per cage under 12 h:12 h light: dark cycles, maintained at a temperate of 18–24 °C and 40–60% humidity. Mice were anesthetized with 3% isoflurane (1.5 L/min oxygen) for 120 s, placed in the injury device still on isoflurane, and impacted at 2.35 m/s with a 7.5 mm impact depth. The time from removal from the anesthesia chamber to initiation of impacts was 60 s. Five successive impacts were delivered. This procedure was conducted daily for 6 days, such that at the culmination of the injury period mice had received 30 impacts. Sham mice received identical handling and anesthesia but no impacts. Total anesthesia time per day is 3 min[14,84].

**Memantine administration.** Mice were randomly assigned in a blinded manner to receive either memantine (M9292, Sigma-Aldrich, St. Louis, MO) or saline, 1 h prior to head impact. Memantine was administered in saline intraperitoneally at 10 mg/kg in 10 mL/kg daily for the 6-day duration of the HF-HI protocol. Route of administration, safety, and efficacy of this dosing regimen in rodents have previously been reported[85,86].

**Behavioral testing.** To identify hippocampal-dependent learning and memory deficits, we used the Barnes maze, Morris Water maze, and T-maze[26,87,88]. The Barnes maze was conducted using a white plastic apparatus (San Diego Instruments) 0.914 m in diameter. The target hole was randomly selected, and four distal visual clues were used to enhance visuospatial learning and memory. The spatial acquisition phase consisted of two 180 s trials per day for 4 consecutive days. A probe trial was conducted 72 h following the acquisition phase. Mice were tracked using AnyMaze tracking system.

The Morris Water maze was conducted using a 1.2 m diameter pool (San Diego Instruments, San Diego, CA). Four distal visual clues were used to enhance visuospatial learning and memory, and a hidden platform (0.1 m in diameter) was submerged 10 mm below the water. The spatial acquisition phase consisted of two 90 s trials per day for 4 consecutive days. A probe trial was conducted at 72 h following the acquisition phase. The mice were tracked using AnyMaze tracking system. A visual probe trial was conducted at the end of the probe trial to confirm the visual and motor ability of the mice to complete the task.

We tested spontaneous alternation in HF-HI mice using a beige mouse T-maze (San Diego Instruments). Mice were placed in the start area and allowed to acclimatize for 30 s. Once the start door was opened, mice were allowed to ambulate up the starting arm and make a choice of the left or right arm. Once inside, the door to the arm was closed and the mouse was confined in the chosen arm for 30 s. Mice were then removed from the maze and returned to their cage while all doors were reset. The test was then repeated. This process was repeated three times on each mouse, with at least 1 h between each session. The number of times that the mice chose the novel arm during the second phase of the trial (spontaneous alternation) was recorded. All testing was performed and analyzed blind to condition.

**RNA isolation, cDNA synthesis, and RT-QPCR.** RNA was extracted from brain tissue using TRIzol® reagent (15596026, Invitrogen). The concentration and purity of the RNA were assessed using a NanoDrop 1000 spectrophotometer (Thermo Scientific). One microgram of RNA was reverse transcribed into complementary DNA (cDNA) using a High-Capacity RNA-to-cDNA Reverse Transcription Kit (4368814, Applied Biosystems, Foster City, CA). The resulting cDNA was diluted 1:3 with diethylpyrocarbonate-treated H$_2$O for use in RT-QPCR.

RT-QPCR was performed in triplicate in standard 384-well plates (4309849, Applied Biosystems) using the Prism 7900HT fast sequence detection system (Applied Biosystems). Taqman probes (Supplemental Methods) were analyzed under the following cycle conditions: 50 °C for 2 min, 95 °C for 20 s, (95 °C for 1 s, 60 °C for 20 s) with 40 cycles. SDS 2.4 software (Applied Biosystems) was used to generate threshold cycle (Ct) values, and fold change (FC) in mRNA expression was calculated using the ΔΔCt method ($2^{-ΔΔCt}$)[89,90].

**RNA-seq and analysis.** RNA was extracted from cortex and hippocampal brain tissue and RNA quality was assessed using an Agilent 2100 Bioanalyzer RNA 6000 Nano Chip (Agilent). cDNA libraries were prepared from ~1 μg total RNA using the TruSeq-Stranded Total RNA Sample Preparation Kit (Illumina, cat. no. RS-122-2301) with Ribo-Zero gold ribosomal RNA beads as per the manufacturer's protocol. Prior to sequencing, cDNA libraries were quantified by digital droplet PCR using a ddPCR Library Quantification Kit (1863040, Bio-Rad). Libraries were multiplexed with six samples per pool and 7 pM of each library was hybridized to an eight-lane flowcell following cluster generation using an Illumina cluster station. Libraries were then sequenced on an Illumina Hi-Seq sequencer using Hiseq v4 chemistry, generating 100 bp paired-end reads.

RNA-seq reads were aligned to the mouse reference genome (mm10) using STAR[91] and expression counts per transcript quantified using eXpress[92]. From the raw counts per transcript, we used DESEQ2[93] to normalize data and estimate differential expression between groups, and generate datasets containing contrasts for HF-HI treatment compared to sham. Briefly, from raw counts our DESeq2 pipeline estimated size factors (to account for differences in library depth) and gene-wise dispersions before shrinking these estimates to generate an accurate dispersion to model the counts. Then, we fit the negative binomial model for differential expression testing to generate contrasts per region and Log 2 FCs (LFCs) for each sample group, before performing hypothesis testing using the Wald test. To improve power detection, outliers of genes whose mean of normalized counts was below the default threshold were removed using Cook's distance cutoffs. To determine significance, the Wald test takes the LFCs and divides it by its standard error, resulting in a $z$-statistic. This $z$-statistic is then compared to a standard normal distribution, in order to compute a $P$ value for each specific gene. To correct for multiple testing and control the FDR, the Benjamini–Hochberg (BH) algorithm was used set to a threshold of 0.1.

Following DESeq2 analysis and generation of variance stabilized contrasts with LFC data, pca, volcano, and dot plots were generated from datasets using ggplot2 and enhanced volcano plot packages [https://github.com/kevinblighe/EnhancedVolcano]. Heatmaps were generated using the pheatmap R package [https://github.com/raivokolde/pheatmap] and Morpheus software. The GWAS Catalog was used to find genes associated with neurodegenerative or cognitive/behavioral traits, and the circos plot including these data was generated using the circlize R package.

For memantine studies, RNA-seq experiments were carried out with the raw counts per transcript normalized using DESEQ2 protocols. For these analyses, interaction terms were added to the design formula such that the two factors of treatment and injury were considered before DESeq2 analysis was performed. Differential expression between groups was analyzed with the model design: Transcript Expression ~ group (Condition × Treatment), where group represents

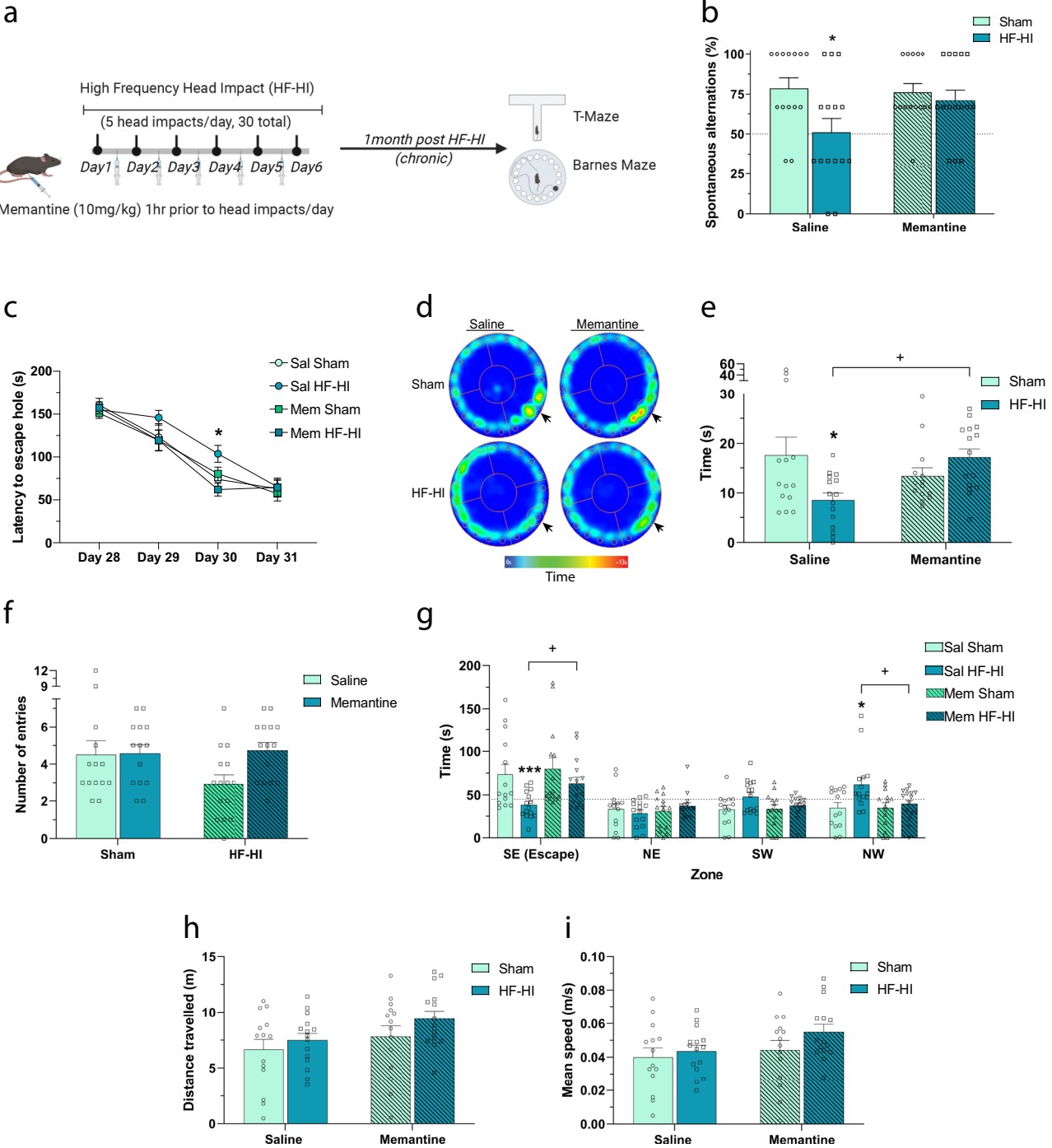

**Fig. 8 The chronic cognitive deficits of HF-HI are prevented by memantine. a** Schematic of the experimental design for this figure. Mice were pretreated each day with memantine 1 h before sham or HF-HI. Cognitive function was tested 1 month after the final impact. **b** Spontaneous alternations for each group during T-maze revealed a significant effect of HF-HI on cognitive function (*$P = 0.013$) that did not occur in memantine-treated mice (sham $n = 14$, HF-HI $n = 15$, two-way ANOVA and Bonferroni post hoc). **c** Latency to the platform during the acquisition phase of the Barnes maze revealed cognitive impairment in saline HF-HI mice, but not in memantine HF-HI mice. Data from two trials per day on four consecutive days (sham $n = 14$, HF-HI $n = 15$, *$P = 0.012$, two-way RM ANOVA and Bonferroni post hoc). **d** Representative heatmaps illustrating the search strategies of each of the groups in the Barnes maze probe trial. **e** Time spent in the escape target during the 72 h delay probe trial was significantly reduced in saline HF-HI mice (*$P = 0.014$) and reversed by memantine ($+P = 0.015$) (sham $n = 14$, HF-HI $n = 15$, two-way ANOVA and Bonferroni post hoc). **f** Number of entries into the escape target during the Barnes maze probe trial. **g** Time spent in each of the quadrants of the Barnes maze was reduced in saline HF-HI mice (***$P = 0.001$) and reversed by memantine ($+P = 0.039$) (sham $n = 14$, HF-HI $n = 15$, two-way ANOVA and Bonferroni post hoc). **h** Distance traveled and **i** mean speed was not changed by treatment group during the probe trial of the Barnes maze (sham $n = 14$, HF-HI $n = 15$). Data are shown as mean ± SEM.

the combined experimental variables of injury and drug. We generated a table containing contrasts for all levels of group, that is, for each treatment in each condition, and then filtered out any genes that showed significant ($P < 0.05$) differential expression using the DESeq2 pipeline.

Furthermore, we performed updated in-depth analysis on European Nucleotide Archive database (ERP015139) deposited data of an RNA-seq dataset from a human transcriptomic study of six late-stage postmortem CTE brains[22]. Evaluation of Gene Ontology term enrichments for both HF-HI and CTE studies was performed using gProfileR within R. Protein interaction networks were acquired from the gProfiler database and visualized using Cytoscape[94]. RNA-seq data that support the findings of this study have been deposited in GEO with the accession code GSE165760.

**Golgi staining and dendritic spine counts**. Golgi staining was performed on hippocampal sections using the FD Rapid Golgi Stain Kit (FD NeuroTechnologies, Ellicott City, MD)[14,63]. Bright-field microscopy images of neurons in the CA1, dentate gyrus, and layer II/III of the cortex were captured by a person blinded to the treatment group, and the number of dendritic spines on apical oblique (AO) or basal shaft (BS) dendrites were quantified. We imaged spines in a 20 μM section of the primary AO dendrites between 30 and 100 μM away from the apical dendrite. Slides and resultant images were coded, and dendritic spines were counted by a blinded rater using the Image J software.

**Long-term potentiation**. Using a Vibratome 3000 Plus (Vibratome, St. Louis, MO), horizontal hippocampal sections (350 μm) were sliced in cutting solution (Supplemental Methods) before slices were incubated for 2 h in artificial cerebrospinal fluid (aCSF) with constant saturation in 95% $O_2$ and 5% $CO_2$. Recordings were performed while slices were maintained in aCSF with 95% $O_2$ and 5% $CO_2$. A unipolar stimulating electrode was placed in the stratum pyramidale layer of hippocampal CA3, and a borosilicate glass electrode filled with 0.9% NaCl internal solution was placed in the stratum radiatum layer of CA1 to record field excitatory postsynaptic potentials. A stimulus input/output curve was generated by exposing the slice to a randomized series of stimulations (10 pulses total for 1 s each, with a 10-s interstimulus interval) that ranged from below the identified threshold to the stimulation that evokes the maximum response and the data were normalized to this maximum response. Using the stimulus that elicited 1/3 of the maximum response, a baseline was recorded for 5 min (10 stimulations total, at 1 s each with a 30-s interstimulus interval). The same stimulus was used in a high-frequency stimulation paradigm to elicit LTP, whereby four tetanic trains were delivered at 100 Hz, 1 s each, with an interstimulus interval of 10 s. Thirty seconds after the tetanic pulses were delivered, LTP was recorded for 1 h, during which stimulation occurred every 30 s (for 1 s) using the same stimulus[95]. Two slices per animals were recorded each day, and their data were normalized between days for comparison.

**Whole-cell patch-clamp recordings**. Twenty-four hours after the last day of impacts, mice were transcardially perfused with ice-cold N-methyl D-glucamine (NMDG) solution[96], before brains extracted and hippocampus sectioned horizontally (300 μm) in ice-cold NMDG bubbled constantly with 95% $O_2$ and 5% $CO_2$. Hemisected slices were then incubated for 12 min each in 32 °C NMDG solution bubbled constantly with 95% $O_2$ and 5% $CO_2$. Individual slices were subsequently left in incubation solution (Supplemental Methods) at room temperature (22–24 °C) for ~4 h prior to recording.

Recordings were done using a constant perfusion (2–3 ml/min) of aCSF2 solution (Supplemental Methods). To record mEPSCs, as well as hyperpolarizing and depolarizing current-clamp recordings from CA1 pyramidal cells, a potassium gluconate (kgluc) internal solution was used (Supplemental Methods). For current-clamp experiments, the current required to hold the cell at ~ −70 mV was determined, and then current injections were performed from there. Measurements of intrinsic properties and the action potential firing were made using the voltage responses to hyperpolarizing and depolarizing current injections (delivered at 20 pA steps for a duration of 400 ms per step). Firing rates in response to depolarizing current injections for each group were determined based on the average action potential firing at each depolarizing step. The rectification was measured from the peak of the voltage response to the hyperpolarizing current injection, and the Ih sag current was extrapolated from the end of the voltage response to the hyperpolarizing current injection. Paired-pulse ratio recordings were performed with kgluc internal solution. To study the paired-pulse ratio, stimulation was delivered ~300 μm from the CA1 pyramidal cells in the stratum radiatum layer (to stimulate Schaffer collateral afferents) using a tungsten bipolar-stimulating electrode (Microprobes for Life Sciences Inc., Gaithersburg, MD). A stimulus that elicited a moderate response was identified, and two pulses were delivered in rapid succession (50 ms apart), while 25 μM BMR was being delivered via y-tube[97–99]. Ten recordings per cell with the stimulus administration were made, and the average ratio of the second pulse to the first pulse was calculated for both groups. Recordings of mEPSC were performed on cells during voltage clamp (at −70 mV), while 1 μM TTX and 25 μM BMR were administered via y-tube to block sodium channels and GABA-mediated inhibitory currents (respectively)[100,101]. For mEPSC amplitude and frequency analyses, templates were

made of stereotypical mEPSC events from a subset of cells in each group using the Clampfit 10.7 software (Molecular Devices LLC, Sunnyvale, CA). These templates were then used to automatically identify and measure the amplitude of each mEPSC for all recordings, and then validated with manual evaluation. For frequency, the total number of events within the same 2-min period for each cell were identified using the template, and the same interval was manually evaluated to confirm the number of mEPSCs. To examine the distribution of mEPSCs, the cumulative distribution function of all events was computed for each cell and averaged across all cells in a group.

Recordings aimed at measuring excitatory input to the CA1 pyramidal cells were performed using a cesium methanesulfonate internal solution[102]. Stimulation was delivered ~300 μm from the CA1 pyramidal cells in the stratum radiatum layer (to stimulate Schaffer collateral afferents) using a tungsten bipolar-stimulating electrode. To assess the AMPA/NMDA ratio, stimulation occurred at an intensity that would evoke EPSCs between 100 and 400 pA in amplitude. First, cells were clamped at −70 mV to estimate the AMPA-mediated currents generated, and the peak amplitudes were recorded. Then, using the same stimulus intensity, the cells were clamped at +40 mV, and the peak amplitude of the current response 60 ms after the stimulus artifact was recorded in order to estimate only the NMDA-mediated contribution. To confirm that the NMDA response was measured at the appropriate time following the stimulus artifact, a subset of cells from both groups was exposed to 5 μM NBQX administered via y-tube in order to eliminate any AMPA-mediated current and isolate the NMDA current. The ratio of the AMPA response to the NMDA response for both the sham and HF-HI groups was then calculated.

To assess the extrasynaptic receptor-mediated contribution to the EPSC, stimulation was delivered ~300 μm from the CA1 pyramidal cells in the stratum radiatum layer (to stimulate Schaffer collateral afferents) using a tungsten bipolar-stimulating electrode. Single stimulations (one every 20 s) were given to elicit a response between 100 and 400 pA at a holding potential of 40 mV (to record NMDA-mediated EPSCs). Then, a high-frequency stimulation was performed (seven stimulations at 200 Hz) to generate glutamatergic "spillover" out of the synapse and engage the extrasynaptic NMDA receptor-mediated response. The decay of the NMDA EPSC was calculated using a double exponential fitting (selected to minimize error). The equation for the double exponential was $f(t) = \sum_{i=1}^{n} A_i e^{-t/\tau_i} + C$, where $C$ was fixed at 0. The values obtained from the double exponential equation (A1, A2, tau1, and tau2) were used to calculated the weighted tau ($\tau_w$), where $\tau_w = tau1*(A1/(A1 + A2)) + tau2*(A2/(A1 + A2))$. The percent charge transfer (%Q) from extrasynaptic NMDA receptors as compared to synaptic NMDA responses to single stimulations were measured for both sham and HF-HI groups. The %Q was calculated by dividing the area under the curve of the extrasynaptic/perisynaptic response by the area under the curve of the response to the single stimulation (both normalized to their own mean amplitude of the peak of the response). Tonic NMDA current was measured in response to stimulations at 40 mV holding potential by subtracting the value of the leak current prior to the addition of (R)-CPP from the value of the leak current after the addition of (R)-CPP. To confirm that the NMDA response was measured at the appropriate time following the stimulus artifact for each of these experiments, a subset of cells from both groups was exposed to 5 μM NBQX administered via y-tube in order to eliminate any AMPA-mediated current and isolate the NMDA current[103].

Recording electrodes made from borosilicate glass were pulled using a vertical heating puller (PP-83, Narishige, Tokyo, Japan). Only those with 3–5 mΩ tip resistance were used for whole-cell recordings. For local field potential recordings, a 1 mΩ tip resistance was used. Patch-clamp recordings were performed using a MultiClamp 700B amplifier, with signals being filtered using a low-pass (2 kHz) filter and sampled at 5 kHz. The signals were acquired using a Digidata 1440A (Molecular Devices, Sunnyvale, CA) on a Dell Optiplex 990 PC.

Electrophysiology data analysis was performed using threshold detection software (Clampfit v10.7, Molecular Devices LLC, Sunnyvale, CA). For local field potential recordings, the stimulus threshold was examined to make sure that the threshold for response of slices varied no >0.1 mA. For all whole-cell patch-clamp experiments, access resistance was monitored for each cell such that they were excluded if a change >20% was observed.

**Statistics and reproducibility**. All data are presented as mean ± SEM. Statistical testing for RNA-seq experiments is discussed in the relevant methods section. All other data were analyzed using GraphPad Prism software (version 8.0). Datasets comparing only two groups were analyzed using unpaired, two-tailed *t* tests. Datasets with more than two groups were analyzed using either one-way ANOVA with Bonferroni post hoc test for single-factor designs, a two-way ANOVA with Bonferroni multiple comparison's post hoc test for two-factor designs, a two-way ANOVA with repeated measures with Bonferroni multiple comparison's post hoc test for two-factor designs with repeated measures from the same subjects. The criteria for statistical significance was preset at $P < 0.05$ for all experiments. The following data in the manuscript were successfully repeated one time with independent cohorts: Barnes maze, Water Maze, T-maze, and mouse RNA-seq. The key-finding data elements (24 h transcriptome data, behavior, and the AMPA/NMDA ratio data) were also successfully replicated in the memantine study, and that data are included in this manuscript.

**Reporting summary**. Further information on research design is available in the Nature Research Reporting Summary linked to this article.

## Data availability

RNA-seq data supporting the findings of this study are deposited in GEO (accession code GSE165760). The statistical analysis for all figures is provided with the paper in Microsoft Excel format. Human CTE RNA-seq data were extracted from the European Nucleotide Archive database (ERP015139) Source data are provided with this paper.

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

## Acknowledgements

These authors contributed equally: Mark R. Cookson, Stefano Vicini, Mark P. Burns. We thank all past and present members of the Burns lab for valuable discussions. This work was supported by the National Institutes of Health (NIH)/National Institute of Neurological Disorders and Stroke (NINDS) (R01NS107370 and UG3NS106941 to M.P.B.), the National Institute of Diabetes and Digestive and Kidney Diseases (NIDDK) (R01DK117508 to S.V.), the Intramural Research Program of the NIH, National Institute on Aging (NIA) (M.R.C.), and the NIA (R03AG061645 to J.-Y.W.). NINDS also supported the Neural Injury and Plasticity Training Grant housed in the Center for Neural Injury and Recovery (CNIR) at Georgetown University (T32NS041218 to S.S.S, C.N.W., and H.T.K). Funding was also provided by the Advanced Rehabilitation Research and Training Fellowship funded by the Department of Health and Human Services (90AR5005 awarded to B.S.M). This research was supported by the Mouse Behavior Core in the Georgetown University Neuroscience Department. Graphical illustrations and schematics throughout the manuscript were made using BioRender (https://biorender.com/). We also thank Ashley Bolte and Ruchelle Buenaventura for their support with bioinformatic coding and visualization of sequencing studies, and also Dr. Patricia Foley, Dr. Robin Tucker, and all of the Division of Comparative Medicine (DCM) team at Georgetown University.

## Author contributions

S.S.S., B.S.M., C.N.W., A.C.H., and H.T.K. performed HF-HI procedures, behavioral testing, and biochemical analysis. D.Z. designed and fabricated the HF-HI pneumatic device. S.S.S., A.P.C., J.-Y.W., J.G.P., and S.V. designed, performed, and analyzed electrophysiology experiments. B.S.M., M.P.B., and M.R.C. designed RNA-seq experiments. B.S.M. and A.K. performed sequencing experiments and B.S.M. and M.R.C. analyzed the data. S.S.S., B.S.M., and M.P.B. analyzed all data and interpreted the results.

## Competing interests

The authors declare no competing interests.
