## [Peer Review File · Nature Communications]

Reviewer #1 (Remarks to the Author):

Understanding how repeated head injuries can lead to the development of long-lasting CNS complications, including neurodegenerative disease, is an important area of study. Here, the researchers are investigating repeated/high frequency head injuries (HF-HI) in mice and the corresponding neurocognitive complications that develop 1 month later. In the paper entitled, "Synaptic adaptation drives cognitive impairment after high- frequency head impact", the authors show that these sub-concussive injuries did not generate inflammation or cell death, but rather affected neuronal plasticity and cognition (1 month later). By comparing their RNAseq data to human CTE RNAseq data they show that the transcriptome signatures identified as being modified by repeated TBI in mice paralleled the neuronal compartments that were degraded in CTE brain. This provides a relevant translational connection of their animal data with human studies of CTE. In addition, the authors show evidence of NMDA receptor activation with injury and mediated alterations in plasticity (e.g, evoked EPSC and LTP). These data seem a bit over interpreted. Last, they used memantine treatment, a non-competitive NMDA receptor antagonist, 1h prior to head impacts on each day of HF-HI, to prevent the stimulation of perisynaptic and extrasynaptic NMDA receptors by glutamate. This prophylactic treatment of mice with memantine reversed the transcriptomic, electrophysiological, and behavioral signature of HF-HI. Overall the authors interpret these data to indicate that behavioral abnormalities after HF-HI can be driven by physiological changes to the brain synapses alone, occurring independently of inflammation, neuronal damage or p-tau accumulation. This idea is supported by their data. They propose that cognitive impairments after HF-HI are caused by activation of synaptic and extra-synaptic NMDA receptors and activation of the immediate early gene *Arc*, resulting in reductions in surface NMDA and AMPA receptors, a net reduction in the AMPA/NMDA ratio, and impaired learning and memory.

1. The RNAseq data, as they are presented, are very hard to interpret. The fonts are so small these data are unreadable. Also, there is no description on how significance was determined for the RNAseq data and the corresponding pathway analyses.
2. For presentation purposes each replicate in its own lane for the heat maps is unnecessary. All the data analyses would be from the group averages. The authors could save a great deal of space, increase font sizes, and present clearer data by showing the average for each treatment group in the heat maps.
3. It is unclear what the directionality of the pathway analyses. There was a balance of genes that were either increased or that were decreased with the repeated head injury. The analyses don't tell us if these correspond pathways are increased or decreased, just that they are significant. One would think that many of these pathways are decreased by injury, but that is never clearly presented in the results or figure legends. The directionality of these pathways and upstream regulators is important in trying to interpret the findings.
4. The statistical approaches are mainly described in the figure legends. But these legends are a bit unclear. For instance, how is figure 1 a 2-way ANOVA? There is only 1 main factor, TBI or sham. If anything, the water maze assessments should be used as a repeated measure ANOVA. When there are 2-way ANOVAs, the main effects of the treatments and the interactions are not provided. For example, what is the main effect of TBI and what is the interaction between memantine (figure 6&7)? The post hoc analyses are unjustified, if there is not a significant interaction between TBI and memantine.

5. In figure 6, which genes are increased by TBI and then reversed by intervention? That isn't clear from the volcano plots or the pathway/regulator analyses. Are all these pathways increased? If half the genes are decreased, how come there is not a suppression of pathways?

6. The electrophysiology data isn't compelling. The effects of TBI at 1 month later are pretty limited (figure 4&5). The primary conclusions of the electrophysiology data are related to AMPA/NMDA ratio. What does a change in this ratio represent? The authors explain that the surface receptor expression of the glutamate receptor have changed, but one does not see how this has been demonstrated in the figures. Also, with all these RNA/pathway data it seems narrow to focus on the authors interpretation of the ephysiology data on just one gene, the Arc gene. The presentation of the data here could be improved to clarify these points.

7. In neurotrauma we always worry about pretreatment of interventions prior to injury. Does post-injury memantine intervention also prevent the transcriptomic, electrophysiological, and behavioral signature of HF-HI? This seems like a relevant point for discussion.

8. It's unclear from the CTE/HI-HF data presented in the paper, but was there an inflammatory component of the CTE RNA seq data in humans? Along with teasing out what is the same, can you determine what is different?

Reviewer #2 (Remarks to the Author):

SUMMARY

The authors describe a series of experiments to analyze the underlying pathophysiology and cognitive deficits associated with a high frequency repetitive traumatic brain injury model. The authors claim that chronic adaptation of the excitatory synapse is the mechanism by which high frequency head impact (30 hits in one week) cause behavioral deficits. They specifically use RNA sequencing, electrophysiological measurements, and immunohistochemistry to assess correlates of cognitive dysfunction they identify with the Barnes maze, Morris water maze, and T-maze behavioral tests. Their transcriptomic profiling suggests robust transcriptional changes associated with synaptic-related gene modules after TBI and their functional studies with electrophysiology also show a decrease in synaptic function, including a reduction in LTP, a reduction in the frequency of mEPSCs, and a reduced AMPA/NMDA receptor ratio with TBI. Little neuropathology is identified including a lack of a significant change in astrogliosis, microgliosis, and white matter damage assessed with APP. Interestingly, pretreatment with memantine, prevents these deficits in cognition, transcriptional changes in synapses, and functional synaptic deficits.

Collectively, these results suggest that there are robust transcriptional changes in response to HF-HI that are enriched for synaptic terms, implying a primarily neuronal contribution to the behavioral phenotypes of this model. However, their conclusions are not supported by the results presented here. The only data they show are acute and there is no direct correlation with the cognitive deficits measured at chronic time points at which time they could very well observe inflammation and other pathologies.

There are numerous fundamental problems such as the disperse timing of many of the experiments,

the lack of rigorous methodological and statistical details and their over interpretation of some of their findings.

Specific concerns are listed as follows:

1. Timing of studies. The timing of all experimental studies needs to be clearly indicated in the methods, results, and figures legends. Often it is only given in one location, and is impossible to follow. Furthermore, sufficiently clear detail is often missing. It seems that behavior was performed one month after repetitive injury, and one assumes this is one month after the last hit deduced from Figure 2. Timing of experiments for all indices measured should be included in a summary pictorial.
2. Further concerns relate to the fact that many measurements including the transcriptomic profiling only occur at an acute time points of 24 hours, while the cognitive deficits are measured at 1 mo after injury. They look at some outcome 24 hr 3 and 7 days post injury and cytokine at 6 hours post injury. There are a lot of data that are not logically developed.
3. They look at some outcome 24 hr 3 and 7 days post injury and cytokine at 6 hours post injury. There are a lot of data that are not logically developed. The transgenic tau animals were only used for a subset of studies They conclude that the HF-HI model is very mild, and does not activate the classical injury cascades or result in the development of commonly observed TBI pathology through the cortex, hippocampus, or corpus callosum, however they only look at acute time points that do not directly relate to the time points when they observe behavioral deficits.
4. From their data they can't conclude that HFHI does not cause inflammatory or classical TBI CTE pathology.
5. The authors should give more details on the rationale for consecutive hits, the specifics of total time of anesthesia per day.
6. The rationale for giving memantine before head impact is unclear as also it has no translatability.
7. All measurements including pathology, electrophysiology, and profiling should really occur at the timing of cognitive deficits. Therefore, additional experiments to measure all indices at both acute and chronic (1 mo) time points are needed to support their conclusions. This especially needs to occur for the synaptic electrophysiology data that showed changes after TBI. Only the decreased AMPA/NMDA ratio after TBI was confirmed at 1 mo, while other notable changes including deficits in LTP and a decreased frequency of mEPSCs were not repeated at 1mo. In addition, they should be repeated for the memantine experiments.
8. Appropriate details regarding electrophysiological analysis are not included in the text. More details are needed than "electrophysiological analysis was performed with threshold detection software". How was the sag current/rectification defined? Was the frequency of firing the average firing frequency or the first firing frequency? Please include the duration of the current step for looking at intrinsic properties. Please detail how the mEPSC analysis was performed, a certain number of events per cell or total events in a specific interval? How were they recorded- presumably at -70 mV in voltage clamp? Where exactly was the stimulus electrode placed for evoked EPSCs and the paired pulse analysis?
9. Many studies are underpowered to draw conclusions. For example, an n=5 is not sufficient for the paired pulse ratio analysis to suggest there are no deficits.
10. The neuropathology should be quantified, not just representative images shown in Supplemental Figure 1.
11. Rigorous statistical detail is completely lacking for this journal in particular. The p value, effect size, degrees of freedom should be given for every statistical test performed.
12. Over interpretation of the data. Much emphasis is placed on the presence of "silent synapses";

however, their data supports a decrease in excitatory synaptic function. less LTP, a decreased frequency of mEPSCs which can also be due to just fewer synapses, and a decrease in the AMPA/NMDA ratio... more careful wording is suggested.

13. The authors add in also transcriptomic of human CTE patients and it is not clear why they add in since the model they claimed has not CTE pathology.

14. There are a lot of independent and unrelated data clumped together with no logic. They claim that the mice have no CTE pathology, yet they use human CTE brain to find common synaptic signature. This link is unclear. The results however further suggest that if they were to look at pathology at chronic time points when they also observe behavioral changes, they might find it.

15. The authors do not have the data to make this conclusion:

“Collectively, these results show that there are robust transcriptional changes in response to HF-HI that are enriched for synaptic terms, implying a primarily neuronal contribution to the behavioral phenotypes of this model. The only data they show are acute and there is no direct correlation with the cognitive deficits measured at chronic time points at which time they could very well observe inflammation and other pathologies.”

Reviewer #3 (Remarks to the Author):

This paper characterizes a correlated set of behavioral, electrophysiological, and gene expression changes driven by a novel paradigm for mild traumatic brain injury, where mice are given a series of 5 sequential mild impacts daily for a period of 6 days to mimic the frequent but low-intensity impacts associated with human athletics. Interestingly, this paradigm produces few if any of the inflammatory and pathological signatures associated with most injury models, but still causes clear behavioral deficits in learning and memory tasks, suggesting that they are instead due to synaptic adaptations rather than other injury-driven biological processes often thought to be important in cognitive deficits. Electrophysiological changes in hippocampus include reduced plasticity, a smaller NMDA component of excitatory transmission, and reduced mEPSC frequency. The synaptic effects are paralleled by robust gene expression changes that are strongly enriched for genes related to synapses. This is an interesting finding for the field, and the integration of a wide range of approaches and measurements is a major strength. I can't comment on gene expression analyses in detail, but the use of multiple behavioral paradigms and multiple electrophysiological approaches add considerable scope. The demonstration of an effective preventive treatment is an additional strength. Overall the manuscript points towards injury-induced synaptic adaptations, distinct from other classical injury responses, as a contributor to cognitive disruptions in mild injury.

Major comments -

Overall the paper shows diverse changes clearly supported by the data. In a few cases the authors draw inferences that are reasonable but not directly demonstrated by the data. One is the strong unequivocal statement about 'formation of silent synapses' in the abstract. This conclusion isn't clearly explained but seems to be based on the fact that (1) mEPSC amplitude is stable but frequency goes down while (2) paired pulse amplitudes are unchanged on average and (3) there is no loss of spines. Silent synapses per se aren't directly demonstrated. The numerous synaptic changes seen are interesting on their own independent of this interpretation.

Similarly in the concluding statement, lines 500-504: 'excitatory synapses are reducing their

glutamatergic receptor densities' : this is a bit vague and seems related to the silent synapse idea. mEPSC amplitude stays similar suggesting that receptor content actually does not change, at least in the synapses contributing to these data. Selective and total loss of AMPA receptors at a specific subset of synapses is one way to reconcile this finding with reduced mEPSC frequency / lack of spine loss / unchanged paired-pulse ratio. But this kind of selectivity would be somewhat surprising and if this interpretation is taken it should be backed up by more direct measurements.

Another is the idea that synaptic adaptation causally 'drives' cognitive impairment as stated in the title and elsewhere. At a minimum synaptic adaptation very likely contributes, but more formally the data show correlated changes in behavior, gene expression, and synaptic transmission (plus excitability). All of these correlated changes are reversed by memantine but this doesn't strictly establish causality. It's still possible that they are each independently driven by some other process. This is not a concern with the findings themselves but a matter of tempering conclusions and language.

Memantine has an impressive effect. The abstract states a 'complete reversal' of all injury effects but not all ephys measurements were repeated for memantine, just AMPA/NMDA ratios. Given the central focus of the paper on synaptic changes it would be nice to see whether it rescues some of the other synaptic changes but if this isn't possible then the language on this point should be softened a bit.

In Figure 2 there is a wealth of interesting data but it is extremely dense and the font is tiny and hard to read which makes it very difficult to interpret. Perhaps consider breaking up this into multiple figures or moving some panels to supplementary information? Figures 3 and 6 are similarly difficult to absorb. Figure 4 would be easier to read if the axis labels were all larger size like 4H/4I.

Minor points -

Fig 2A,B – 'in terms of overall cortical gene expression'
– would be nice to have labels indicating units for scale bars

Lines 452-454: 'peak amplitude of the field excitatory postsynaptic potential (fEPSP) is shifted downward, indicating that LTP is significantly reduced compared to sham mice'
- phrasing here is unclear. Does 'downward' mean that the normalized amplitude is smaller than in controls after LTP induction, so that strength is potentiated to a lesser degree in HF-HI?
- LTP is often quantified using the slope of the fEPSP. This probably gives a similar answer, but it could be nice to know if effects are comparable for comparison with other work.

Fig. 4B – what were the peak amplitudes normalized to? Normalization is especially important for these input-output experiments and should be clearly stated.

Fig. 4C – are the zoomed-in images stretched horizontally somehow? They seem elongated relative to the lower-mag images. Scale bars would be helpful here.

Fig. 4F – there is no statistical change in average paired-pulse ratio, but the variance of this measure does seem to increase. Could this be a real effect of injury, with different neurons affected in different directions?

Fig S4C (mEPSC measurements) - these are an important counterpart to the PPR and silent synapse argument, but are harder to access in the supplemental materials. The data in S4A/B/C could be useful to show in the main figure panel especially since there is such a strong shift in mEPSC frequency.

Line 475-476: 'despite there being no difference between the sham and HF-HI groups in approximate rheobase following depolarizing current injections (Fig. S4D)':

- is rheobase really identical? This seems somewhat surprising given the apparent difference in firing thresholds seen in the current-frequency plot in Fig 5B, and the examples in 5A/5C.

Line 476-477: This acute change in neuron excitability had normalized at 1m after HF-HI (Fig.5C-D).

- the groups are indistinguishable here in 5D, but the curves also look different than either group in 5C (that is they show a lower threshold). Can the authors comment on possible reasons for this?

Line 488-490: 'To determine whether changes in the relative expression or conductance of extrasynaptic NMDA receptors could be contributing to the functional changes that we observe'

- what changes are the authors referring to? to the altered firing and reduced excitability described immediately beforehand? or the changes in plasticity earlier on? Both? The results jump from synaptic measurements, to firing properties, back to synaptic measurements. It might be easier to follow if the synaptic changes were all reported together – unless the authors are proposing that extrasynaptic NMDARs are responsible for both?

Line 498-499: 'Additionally, HF-HI does not appear to change glial glutamate transporter capabilities.'

- is this inferred from the NMDAR current decay? If so this should be stated, otherwise this statement implies transporter function were measured directly (e.g. with astrocyte recordings).

- 'with a net reduction in the AMPA/NMDA ratio which, along with changes in intrinsic excitability, drive acute and chronic plasticity impairments' – similar to above, these are likely to contribute but the causal relationship is not directly shown. The data here don't specifically establish whether plasticity effects arise from one or both of these changes. There could be contributions from other changes in synaptic protein content, signaling pathways, etc. Language should be more careful.

Lines 346-349: In addition, Iba1, CD68 and GFAP staining reveal no changes in microglial or astrocyte activation/morphology in the cortex, hippocampus or corpus callosum of HF-HI mice compared to sham counterparts at 1d, 3d, or 7d post-HFHI (Fig. S1A,D-F)

- lack of classical neuroinflammatory markers seems to be a major point of the study. The panels show single qualitative examples to support this idea. How was this evaluated or quantified for consistency? The legend refers to 'n = 6 per group' but it's not clear if this applies to both immunostaining and biochemistry?

Reviewer #1 (Remarks to the Author):

Understanding how repeated head injuries can lead to the development of long-lasting CNS complications, including neurodegenerative disease, is an important area of study. Here, the researchers are investigating repeated/high frequency head injuries (HF-HI) in mice and the corresponding neurocognitive complications that develop 1 month later. In the paper entitled, “Synaptic adaptation drives cognitive impairment after high- frequency head impact”, the authors show that these sub-concussive injuries did not generate inflammation or cell death, but rather affected neuronal plasticity and cognition (1 month later). By comparing their RNAseq data to human CTE RNAseq data they show that the transcriptome signatures identified as being modified by repeated TBI in mice paralleled the neuronal compartments that were degraded in CTE brain. This provides a relevant translational connection of their animal data with human studies of CTE. In addition, the authors show evidence of NMDA receptor activation with injury and mediated alterations in plasticity (e.g, evoked EPSC and LTP). These data seem a bit over interpreted. Last, they used memantine treatment, a non-competitive NMDA receptor antagonist, 1h prior to head impacts on each day of HF-HI, to prevent the stimulation of perisynaptic and extrasynaptic NMDA receptors by glutamate. This prophylactic treatment of mice with memantine reversed the transcriptomic, electrophysiological, and behavioral signature of HF-HI.

Overall the authors interpret these data to indicate that behavioral abnormalities after HF-HI can be driven by physiological changes to the brain synapses alone, occurring independently of inflammation, neuronal damage or p-tau accumulation. This idea is supported by their data. They propose that cognitive impairments after HF-HI are caused by activation of synaptic and extra-synaptic NMDA receptors and activation of the immediate early gene Arc, resulting in reductions in surface NMDA and AMPA receptors, a net reduction in the AMPA/NMDA ratio, and impaired learning and memory.

1. The RNAseq data, as they are presented, are very hard to interpret. The fonts are so small these data are unreadable. Also, there is no description on how significance was determined for the RNAseq data and the corresponding pathway analyses.

Firstly, we thank the reviewer for their positive comments and highlighting that ***“understanding how repeated head injuries can lead to the development of long-lasting CNS complications, including neurodegenerative disease, is an important area of study.”*** We also agree that our RNA sequencing approaches and identification of modified neuronal transcriptional signatures in both repeated TBI and degraded CTE brain provides relevant translational connection of our animal data with human studies of CTE.

We fully acknowledge that the volume of data we have collected over 3 independent RNAseq studies (24h, 1m & Saline/Memantine treatment paradigms) makes efficient presentation of data difficult with space constraints. To make interpretation clearer, we have significantly restructured our RNAseq figures, removing some analysis and moving other analysis (cytoscape) into supplemental data to allow for more space to increase both font and graph size.

We have also added the following detail to the methods section outlining how significance was determined for RNAseq datasets:

“Briefly, from raw counts our DESeq2 pipeline estimated size factors (to account for differences in library depth) and gene-wise dispersions before shrinking these estimates to generate an accurate dispersion to model the counts. Then we fit the negative binomial model for differential expression testing to generate contrasts per region and Log2 foldchanges (LFC) for each sample group, before performing hypothesis testing using the Wald test. To improve power detection, outliers of genes whose mean of normalized counts was below the default threshold were removed using Cook’s distance cutoffs. To determine significance, the Wald test takes the LFC and divides it by its standard error, resulting in a z-statistic. This z-statistic is then compared to a standard normal distribution, in order to compute a p-value for each specific gene. To correct for multiple testing and control the false discovery rate (FDR), the Benjamini-Hochberg (BH) algorithm was used set to a threshold of 0.1.

Following DESeq2 analysis and generation of variance stabilized contrasts with Log2 fold change data, pca, volcano and dot plots were generated from datasets using ggplot2 [<https://ggplot2.tidyverse.org>] and enhanced volcano plot packages [<https://github.com/kevinblighe/EnhancedVolcano>]. Heatmaps were generated using the pheatmap R package [<https://github.com/raivokolde/pheatmap>], and Morpheus software [<https://software.broadinstitute.org/morpheus>]. The GWAS Catalog was used to find genes associated with neurodegenerative or cognitive/behavioral traits [<https://www.ebi.ac.uk/gwas/home>], and the circos plot including these data was generated using the circlize R package.

*For Memantine studies, RNA sequencing experiments were carried out with the raw counts per transcript normalized using DESEQ2 protocols as described above. For these analyses, interaction terms were added to the design formula such that the two factors of treatment and injury were considered before DESeq2 analysis was performed. Differential expression between groups was analyzed with the model design: Transcript Expression~group (Condition*Treatment), where group represents the combined experimental variables of injury and drug. We generated a table containing contrasts for all levels of group, ie for each treatment in each condition, then filtered out any genes that showed significant ($p < 0.05$) differential expression using the DESeq2 pipeline as described above.”*

2. For presentation purposes each replicate in its own lane for the heat maps is unnecessary. All the data analyses would be from the group averages. The authors could save a great deal of space, increase font sizes, and present clearer data by showing the average for each treatment group in the heat maps.

While we understand the need for space, clarity and increased font sizes, we believe it is important to show individual animal replicates in heatmaps containing only 2 groups (HF-HI v Sham, at 24h & 1m). We feel it is important to provide the reader with a full view to visually assess the results and enable the reader to observe the trends of expression for genes across injury conditions. It also allows the reader to visually interpret the intragroup variance in identified gene clusters.

In saying this, we have taken the reviewers recommendation on board for the complex four-way analysis in the memantine study so we do not present data from 36 individual mice. We now present grouped analysis in that figure (Fig 7 and Fig S6). We have also restructured our other RNAseq figures as described above to decompress the volume of data and allow more space for increased font size in Figures 1,4,5 & 7. In addition, we have added dot plots graphs to more clearly visualize Gene Ontology directionality data in Figures 1, 4 & 7 and S6.

3. It is unclear what the directionality of the pathway analyses. There was a balance of genes that were either increased or that were decreased with the repeated head injury. The analyses don't tell us if these correspond pathways are increased or decreased, just that they are significant. One would think that many of these pathways are decreased by injury, but that is never clearly presented in the results or figure legends. The directionality of these pathways and upstream regulators is important in trying to interpret the findings.

In terms of directionality, it is correct that there is a balance of genes that are increased and decreased in response to head impact. We have now made this clearer with large heatmaps demonstrating top upregulated and downregulated genes for each specific Gene Ontology (GO) classifier (Fig 1G/H, Fig 4D/E/G, Fig 7I, Supp Figs 3&4). We have added new dot plot analysis to visualize the directionality, gene size and gene ratio of key Cellular Component and Biological Process GO pathways (Figs 1,4 &7, S6). Additionally, supplemental tables have been updated to provide information on upregulated and down regulated genes for each RNAseq experiment, as well as information on the directionality of Gene Ontology pathway analysis.

The reviewer has hit on one of the key aspects of this data that we find most intriguing – the synaptic pathways are not simply downregulated – they are being modified. Because we are dealing with synaptic modification, and not synaptic loss, we find that pathways such as “synaptic signaling” are identified as both a significantly upregulated pathway and a significantly downregulated pathway.

4. The statistical approaches are mainly described in the figure legends. But these legends are a bit unclear. For instance, how is figure 1 a 2-way ANOVA? There is only 1 main factor, TBI or sham. If anything, the water maze assessments should be used as a repeated measure ANOVA. When there are 2-way ANOVAs, the main effects of the treatments and the interactions are not provided. For example, what is the main effect of TBI and what is the interaction between memantine (figure 6&7)? The post hoc analyses are unjustified, if there is not a significant interaction between TBI and memantine.

The old Fig 1 had multiple different analysis, including 2-way RM ANOVAs for the acquisition trials of the water maze and Barnes maze. To improve clarity, we have added a comprehensive table to supplemental data which gives the raw data of the group means for each experiment, exact n numbers used in each data panel presented, t-values, degrees of freedom (df), f-values, exact p-values and the analysis used for all data presented in the manuscript. F-values and DF are also now included in the

results section, and we have put statistical data in the figure legend within each panel description, and not just at the end of the figure legend.

5. In figure 6, which genes are increased by TBI and then reversed by intervention? That isn't clear from the volcano plots or the pathway/regulator analyses. Are all these pathways increased? If half the genes are decreased, how come there is not a suppression of pathways?

We have included new analysis in our memantine transcriptome data (now Figure 7) to clearly show how memantine affects key synaptic pathways. We show that memantine HFHI mice have less than half the number of genes affected by head impact as vehicle treated mice (676 genes in saline HFHI vs 338 in memantine HFHI mice), with only 68 being overlapping genes. We have also added a heatmap to show the effect of memantine on the top 20 up and downregulated genes identified in the saline HF-HI group (Fig. 7G). We have highlighted the genes involved in the glutamatergic synapse and show that they are reversed by memantine pretreatment. Pathway enrichment shows that there is a strong attenuation of the HFHI synaptic pathways by memantine (Fig 7H).

In addition, supplemental data is included in Fig S6 and supplemental tables have been updated to provide Log2FC information on the significantly upregulated and down regulated genes after HF-HI, and a comparison of how memantine influences the expression of these genes.

6. The electrophysiology data isn't compelling. The effects of TBI at 1 month later are pretty limited (figure 4&5). The primary conclusions of the electrophysiology data are related to AMPA/NMDA ratio. What does a change in this ratio represent? The authors explain that the surface receptor expression of the glutamate receptor have changed, but one does not see how this has been demonstrated in the figures. Also, with all these RNA/pathway data it seems narrow to focus on the authors interpretation of the ephysiology data on just one gene, the Arc gene. The presentation of the data here could be improved to clarify these points.

In brief, a reduced AMPA/NMDA ratio is indicative of a reduced basal strength of the synapse.

(Excerpt from Kauer & Malenka, 2007 - Nature Reviews Neuroscience 8, 844-858): The basal strength of excitatory synapses is difficult to compare between different cells and preparations. Calculating the ratio of AMPAR-mediated synaptic currents to NMDAR-mediated synaptic currents of a population of stimulated synapses is a normalization procedure that facilitates such comparisons because it is independent of parameters such as the positioning of electrodes or the number of synapses that are activated.

A reduction in the AMPA/NMDA ratio, in the absence of a loss in synapses, and with a reduction in mEPSC frequency (but not amplitude) is considered a signature of a silent synapse, which is where our original interpretation of silent synapses being present is derived from. These silent synapses are extraordinarily difficult to directly measure in hippocampal neurons, which is why we extrapolated our data to this conclusion. In our resubmission, we now note that mEPSC in HF-HI mice have returned to sham frequency by 1m, so in light of this new data we are stepping back from our original interpretation.

We have also added more data to the 1m timepoint, including LTP, neuron excitability, and mEPSC. From our patch clamp data, we find that only the AMPA/NMDA ratio is still reduced at 1m, as is LTP. HFHI mEPSC and neuron excitability have returned to sham levels by 1m, which shows that the changes in these neuron properties were acute impact-related events. We believe that the acute change in excitability was due to ionic imbalance at the time of head impact, and was driven by an acute change in inward potassium channel rectification. This is resolved by 1m post-impact, showing that the decrease in excitability is not driving the chronic memory issues. The excitability/potassium channel data is a novel finding in the field of TBI, as this is the first evidence showing that inward rectifying potassium channels are functionally impaired after any mTBI (or any experimental TBI).

We originally focused on *Arc* as it is such a key mediator of AMPA-mediated memory formation and our electrophysiology data really zones in on a reduction in the AMPA ratio as a key acute and chronic issue in our mice. *Arc* is one of the top 3 genes affected in both 1d and 1m RNAseq analysis, and we have found that approximately 70% of the genes downstream of *Arc* are also affected in our mice (not shown). However, we agree that the focus on *Arc* is somewhat premature and we have altered our results and discussion with less of a focus on *Arc*.

A final word about the electrophysiology data is that it is a subtle, but chronic effect that remains at 1m post-trauma. We never expected to find a very strong effect on synaptic function effect in our HF-HI model. It is supposed to be a very, very minor head impact. But the subtle consequence of a high frequency of these events is long-lasting in the mouse brain.

7. In neurotrauma we always worry about pretreatment of interventions prior to injury. Does post-injury memantine intervention also prevent the transcriptomic, electrophysiological, and behavioral signature of HF-HI? This seems like a relevant point for discussion.

We used memantine as a “proof-of-concept” drug. We believe that each impact causes a release of glutamate from the presynaptic neuron that reaches high enough quantities to stimulate extra-synaptic NMDA receptors. The use of memantine to block downstream changes is our mechanistic link to show how HFHI mediates downstream synaptic modifications and memory deficits.

This paper will establish synaptic modification as an event that occurs after repeated head impact, and then we can turn to discovering treatment strategies to unlock synaptic potential use in affected individuals who are in the chronic phase after the cessation of head impact. Memantine will not be a treatment option for these individuals, as we have shown that the synaptic modifications occur in the acute phase, with only key elements remaining through the chronic phase.

We have increased our discussion of this point.

8. It's unclear from the CTE/HI-HF data presented in the paper, but was there an inflammatory component of the CTE RNA seq data in humans? Along with teasing out what is the same, can you determine what is different?

Firstly, we thank the reviewer for their comments highlighting the importance and value of our translational connection ***“By comparing their RNAseq data to human CTE RNAseq data they show that the transcriptome signatures identified as being modified by repeated TBI in mice paralleled the neuronal compartments that were degraded in CTE brain. This provides a relevant translational connection of their animal data with human studies of CTE.”***

We agree with the reviewers comment and have expanded our analysis further to tease apart key genetic signatures and interactions between our animal model and the CTE brain.

We have added analysis of all overlapping differentially expressed genes (DEGs, Fig 5A), and then used a database of genome-wide association studies (GWAS Catalog), to find known gene associations that have been identified with neurodegenerative or cognitive/behavioral traits. From this we have identified 61 genes conserved between HF-HI model and the CTE brain that are associated with these various neurological disease states and behavioral phenotypes (Fig 5B).

As well as reassessing all DEGs as a whole, we then reanalyzed the WGCNA defined “neuronal” (Fig 5D), and “inflammation” (Fig 5F) clusters as originally identified by Seo, J. S. *et al. Exp Mol Med*, doi:10.1038/emm.2017.56 (2017). While we find zero overlap with the inflammatory component (Fig 5G), we find that the neuronal overlapping genes between 1d & 1m HFHI mice, and CTE brain are genes

involved in neurodegeneration and glutamatergic synaptic processes (Fig 5E). Finally, we identified 19 overlapping genes involved in synaptic signaling BP in all 3 groups (Fig S5). We have expanded on our discussion of these findings.

In terms of determining what is different, we have now included supplemental tables with extensive analysis of all gene comparisons. This table not only provides information of the genes that overlap in all groups, it also provides lists of genes that do not overlap, or are only conserved in 2 groups.

Reviewer #2 (Remarks to the Author):

SUMMARY

The authors describe a series of experiments to analyze the underlying pathophysiology and cognitive deficits associated with a high frequency repetitive traumatic brain injury model. The authors claim that chronic adaptation of the excitatory synapse is the mechanism by which high frequency head impact (30 hits in one week) cause behavioral deficits. They specifically use RNA sequencing, electrophysiological measurements, and immunohistochemistry to assess correlates of cognitive dysfunction they identify with the Barnes maze, Morris water maze, and T-maze behavioral tests. Their transcriptomic profiling suggests robust transcriptional changes associated with synaptic-related gene modules after TBI and their functional studies with electrophysiology also show a decrease in synaptic function, including a reduction in LTP, a reduction in the frequency of mEPSCs, and a reduced AMPA/NMDA receptor ratio with TBI. Little neuropathology is identified including a lack of a significant change in astrogliosis, microgliosis, and white matter damage assessed with APP. Interestingly, pretreatment with memantine, prevents these deficits in cognition, transcriptional changes in synapses, and functional synaptic deficits.

Collectively, these results suggest that there are robust transcriptional changes in response to HF-HI that are enriched for synaptic terms, implying a primarily neuronal contribution to the behavioral phenotypes of this model. However, their conclusions are not supported by the results presented here. The only data they show are acute and there is no direct correlation with the cognitive deficits measured at chronic time points at which time they could very well observe inflammation and other pathologies.

There are numerous fundamental problems such as the disperse timing of many of the experiments, the lack of rigorous methodological and statistical details and their over interpretation of some of their findings.

Specific concerns are listed as follows:

1. Timing of studies. The timing of all experimental studies needs to be clearly indicated in the methods, results, and figures legends. Often it is only given in one location, and is impossible to follow. Furthermore, sufficiently clear detail is often missing. It seems that behavior was performed one month after repetitive injury, and one assumes this is one month after the last hit deduced from Figure 2. Timing of experiments for all indices measured should be included in a summary pictorial.

&

2. Further concerns relate to the fact that many measurements including the transcriptomic profiling only occur at an acute time points of 24 hours, while the cognitive deficits are measured at 1 mo after injury. They look at some outcome 24 hr 3 and 7 days post injury and cytokine at 6 hours post injury. There are a lot of data that are not logically developed.

&

3. They look at some outcome 24 hr 3 and 7 days post injury and cytokine at 6 hours post injury. There are a lot of data that are not logically developed. The transgenic tau animals were only used for a subset of studies. They conclude that the HF-HI model is very mild, and does not activate the classical injury cascades or result in the development of commonly observed TBI pathology through the cortex, hippocampus, or corpus callosum, however they only look at acute time points that do not directly relate to the time points when they observe behavioral deficits.

&

7. All measurements including pathology, electrophysiology, and profiling should really occur at the timing of cognitive deficits. Therefore, additional experiments to measure all indices at both acute and chronic (1 mo) time points are needed to support their conclusions. This especially needs to occur for the synaptic electrophysiology data that showed changes after TBI. Only the decreased AMPA/NMDA ratio after TBI was confirmed at 1 mo, while other notable changes including deficits in LTP and a decreased frequency of mEPSCs were not repeated at 1mo. In addition, they should be repeated for the memantine experiments.

We are now focusing only on 1d and 1m data to better fit out transcriptome and electrophysiology data and to streamline our manuscript. We have added schematics into the figures to demonstrate what data are being collected at each timepoint, and we have removed data that falls outside of the 1d and 1m timepoints.

We have also included the following new data to address the reviewers concerns and to show what synaptic signature events are acute events and what signature remains in the chronic phase:

New Data at 1d:

- T-maze behavioral data
- Directionality data to RNAseq analysis
- Quantification of Iba1 and CD68 immunohistochemistry in the hippocampus, corpus callosum, and optic tract
- Additional data in the paired pulse ratio analysis.

New Data at 1m:

- RNAseq analysis
- Comparison of CTE brain to 1m and 1d HFHI brain
- LTP
- Paired Pulse ratio
- mEPSC
- Neuron excitability data
- Inward potassium channel rectification
- Quantification of Iba1 and CD68 in the hippocampus, corpus callosum, and optic tract
- Cytokine ELISA for TNF α and IL1 β in cortex and hippocampus
- Western blots for Iba1
- PCR for Iba1 and GFAP mRNA
- P-tau and A β data from the cortex and hippocampus at 1m

New Memantine data:

- Modified RNAseq analysis to show directionality and study the effect of memantine on impacted pathways.

4. From their data they can't conclude that HFHI does not cause inflammatory or classical TBI CTE pathology.

We have now included 1m timepoints with Iba1 and CD68 quantification to demonstrate the lack of inflammatory pathology at this chronic timepoint outside of the optic tract. We have also included Iba1 western blots, Iba1 and GFAP mRNA, cytokine ELISAs, tau Western blots and Abeta ELISAs at the 1m timepoint. In addition, transcriptome analysis at 1d and 1m does not tag inflammatory pathways as playing a role in the acute or chronic sequelae of HF-HI.

5. The authors should give more details on the rationale for consecutive hits, the specifics of total time of anesthesia per day.

We have included this rationale in the manuscript, and given the anesthetic specifics (each sham and HFHI mouse receives 3 minutes of 3% isoflurane per day for 6 consecutive days).

The rationale for thirty hits over the space of a week is because the average number of impacts a college football player receives in a week is 21, with defensive ends receiving 41 head impacts per week (Crisco et al 2010). We give these head impacts consecutively, as many of the head impacts can be received in an individual play when a player is tackled to the ground.

We designed this model to be very minor and deliberately picked an impact level that did not produce pathology. When we give this impact to an un-anesthetized mouse, there is no loss of consciousness and we believe it is equivalent to a minor sub-concussive blow to the head.

6. The rationale for giving memantine before head impact is unclear as also it has no translatability.

We used memantine as a "proof-of-concept" drug to show the mechanism of action of HF-HI. We believe that each impact causes a release of glutamate from the presynaptic neuron that reaches high enough quantities to stimulate extra-synaptic NMDA receptors. The use of memantine to block downstream changes is our mechanistic link to show how HFHI mediates downstream synaptic modifications and memory deficits.

This paper will establish synaptic modification as an event that occurs after repeated head impact, and then we can turn to discovering treatment strategies to unlock synaptic potential use in affected individuals who are in the chronic phase after the cessation of head impact. Memantine will not be a treatment option for these individuals, as we know that the synaptic modifications occur in the acute phase, with only key elements remaining through the chronic phase.

We have increased our discussion of this point.

8. Appropriate details regarding electrophysiological analysis are not included in the text. More details are needed than "electrophysiological analysis was performed with threshold detection software". How was the sag current/rectification defined? Was the frequency of firing the average firing frequency or the first firing frequency? Please include the duration of the current step for looking at intrinsic properties. Please detail how the mEPSC analysis was performed, a certain number of events per cell or total events in a specific interval? How were they recorded-presumably at -70 mV in voltage clamp? Where exactly was the stimulus electrode placed for evoked EPSCs and the paired pulse analysis?

We apologize for the oversight. We have included this information in the methods.

9. Many studies are underpowered to draw conclusions. For example, an n=5 is not sufficient for the paired pulse ratio analysis to suggest there are no deficits.

We have increased the n of the PPR analysis at 24h to 8 sham and 10 HFHI cells. It should be noted that these are not cells from just a few mice, but are individual cells from 8 and 10 individual mice to increase rigor. We have also introduced a new figure for 1m PPR analysis with a n = 9 for each group from 9 individual animals in each group.

We have included all information on n for each graph in a table in the Supplemental material.

0. The neuropathology should be quantified, not just representative images shown in Supplemental Figure 1.

We are now only showing 1d and 1m pathology, and have quantified this. We have also included tau and Abeta analysis at 1m, cytokine assays at 1m, Iba1 and GFAP mRNA at 1m, and Iba1 Western blots at 1m.

10. Rigorous statistical detail is completely lacking for this journal in particular. The p value, effect size, degrees of freedom should be given for every statistical test performed.

To improve clarity, we have added a comprehensive table to supplemental data which gives the raw data of the group means for each experiment, exact n numbers used in each data panel presented, t-values, degrees of freedom (df), f-values, exact p-values and the analysis used for all data presented in the manuscript. F-values and DF are also now included in the results section, and we have put statistical data in the figure legend within each panel description, and not just at the end of the figure legend. We have also added detail to the methods section outlining how significance was determined for RNAseq datasets, and this is discussed above in a comment to reviewer 1.

1. Over interpretation of the data. Much emphasis is placed on the presence of “silent synapses”; however, their data supports a decrease in excitatory synaptic function. less LTP, a decreased frequency of mEPSCs which can also be due to just fewer synapses, and a decrease in the AMPA/NMDA ratio... more careful wording is suggested.

Our original interpretation is based on the fact that we have many of the features of a silent synapse in the absence of a loss in the number of synapses. However, with our expansion of our data to 1m, we no longer see a decrease in mEPSC frequency – we only see a change in the AMPA:NMDA ratio. As such, we have modified our interpretation in line with the reviewer’s suggestions.

11. The authors add in also transcriptomic of human CTE patients and it is not clear why they add in since the model they claimed has not CTE pathology.

&

12. There are a lot of independent and unrelated data clumped together with no logic. They claim that the mice have no CTE pathology, yet they use human CTE brain to find common synaptic

signature. This link is unclear. The results however further suggest that if they were to look at pathology at chronic time points when they also observe behavioral changes, they might find it.

The overall objective of this manuscript is to show that synaptic modifications can result in chronic cognitive deficits. The purpose of this research is to fill a knowledge gap between the minimal pathology that is seen in early stage CTE and the strong cognitive deficits that have been reported. We believe that synaptic changes are occurring in the CTE brain, and may occur even earlier than pathological changes.

By comparing our model to the CTE brain, we can see if there is any overlap between a head impact model without pathology, and the diseased brain. We find strong overlap in the synaptic space. However, it also provides us an opportunity to contrast our model to CTE, and HFHI mice do not share the inflammatory component of CTE. While our synaptic modifications are not an exact match for the synaptic downregulation of late-stage CTE, we are encouraged by the fact that both HFHI and CTE are targeting a similar synaptic domain, with both causing similar cognitive dysfunction.

We have now included pathology from 1m HFHI mice.

15. The authors do not have the data to make this conclusion: “Collectively, these results show that there are robust transcriptional changes in response to HF-HI that are enriched for synaptic terms, implying a primarily neuronal contribution to the behavioral phenotypes of this model.” The only data they show are acute and there is no direct correlation with the cognitive deficits measured at chronic time points at which time they could very well observe inflammation and other pathologies.

As requested, we have added extensive 1m ephys and transcriptome data to the manuscript comprehensively concluding that key components of the synaptic adaptation signature we find at 1d remain at 1m. We believe that the use of many different techniques in the acute and chronic phase, plus a mechanistic drug study, that all point to chronic synaptic adaptation in HF-HI mice validates our conclusion.

Reviewer #3 (Remarks to the Author):

This paper characterizes a correlated set of behavioral, electrophysiological, and gene expression changes driven by a novel paradigm for mild traumatic brain injury, where mice are given a series of 5 sequential mild impacts daily for a period of 6 days to mimic the frequent but low-intensity impacts associated with human athletics. Interestingly, this paradigm produces few if any of the inflammatory and pathological signatures associated with most injury models, but still causes clear behavioral deficits in learning and memory tasks, suggesting that they are instead due to synaptic adaptations rather than other injury-driven biological processes often thought to be important in cognitive deficits. Electrophysiological changes in hippocampus include reduced plasticity, a smaller NMDA component of excitatory transmission, and reduced mEPSC frequency. The synaptic effects are paralleled by robust gene expression changes that are strongly enriched for genes related to synapses. This is an interesting finding for the field, and the integration of a wide range of approaches and measurements is a major strength. I can't comment on gene expression analyses in detail, but the use of multiple behavioral paradigms and multiple electrophysiological approaches add considerable scope. The demonstration of an effective preventive treatment is an additional strength. Overall the manuscript points towards injury-induced synaptic adaptations, distinct from other classical injury responses, as a contributor to cognitive disruptions in mild injury.

Major comments -

Overall the paper shows diverse changes clearly supported by the data. In a few cases the authors draw inferences that are reasonable but not directly demonstrated by the data. One is the strong unequivocal statement about ‘formation of silent synapses’ in the abstract. This conclusion isn’t clearly explained but seems to be based on the fact that (1) mEPSC amplitude is stable but frequency goes down while (2) paired pulse amplitudes are unchanged on average and (3) there is no loss of spines. Silent synapses per se aren’t directly demonstrated. The numerous synaptic changes seen are interesting on their own independent of this interpretation.

Similarly in the concluding statement, lines 500-504: ‘excitatory synapses are reducing their glutamatergic receptor densities’ : this is a bit vague and seems related to the silent synapse idea. mEPSC amplitude stays similar suggesting that receptor content actually does not change, at least in the synapses contributing to these data. Selective and total loss of AMPA receptors at a specific subset of synapses is one way to reconcile this finding with reduced mEPSC frequency / lack of spine loss / unchanged paired-pulse ratio. But this kind of selectivity would be somewhat surprising and if this interpretation is taken it should be backed up by more direct measurements.

A reduction in the AMPA/NMDA ratio, in the absence of a loss in synapses, and with a reduction in mEPSC frequency (but not amplitude) is considered a signature of a silent synapse, which is where our original interpretation of silent synapses being present is derived from. These silent synapses are extraordinarily difficult to directly measure in hippocampal neurons (rather than the striatum, where the silent synapse field originates), which is why we extrapolated our data to this conclusion. The chronically high levels of Arc mRNA at 1d and even more so at 1m also helped drive our original conclusion that AMPA receptors are being sequestered away from the synapse surface. However, given the lack of change in mEPSC frequency at 1m, and the feedback from reviewers, we have now modified our conclusions away from silent synapses.

We have also added new data for the 1m timepoint, including LTP, neuron excitability, mEPSC, iEPSC. From our patch clamp data, we find that only the AMPA/NMDA ratio is still reduced at 1m, as is LTP. HFHI mEPSC and neuron excitability have returned to sham levels by 1m, which shows that the changes in these neuron properties were acute impact-related events. We believe that the acute change in excitability was due to ionic imbalance at the time of head impact, and was driven by an acute change in inward potassium channel rectification. This is resolved by 1m post-impact, showing that the decrease in excitability is not driving the chronic memory issues.

Another is the idea that synaptic adaptation causally ‘drives’ cognitive impairment as stated in the title and elsewhere. At a minimum synaptic adaptation very likely contributes, but more formally the data show correlated changes in behavior, gene expression, and synaptic transmission (plus excitability). All of these correlated changes are reversed by memantine but this doesn’t strictly establish causality. It’s still possible that they are each independently driven by some other process. This is not a concern with the findings themselves but a matter of tempering conclusions and language.

Thank you for the comment. We have modified our conclusion and language, including the title.

Memantine has an impressive effect. The abstract states a ‘complete reversal’ of all injury effects but not all ephys measurements were repeated for memantine, just AMPA/NMDA ratios. Given the central focus of the paper on synaptic changes it would be nice to see whether it rescues some of the other synaptic changes but if this isn’t possible then the language on this point should be softened a bit.

We have now included data in our HFHI mice to show that only the reduced AMPA/NMDA remains at 1m post-HFHI. As such, the consistent acute and chronic changes in our model are the transcriptomic

changes, reduced AMPA/NMDA ratio, and altered cognition. We focused on these outcome measures for our memantine studies, as we determined that these are the key events. We have changed our abstract to more accurately reflect this targeted approach.

In Figure 2 there is a wealth of interesting data but it is extremely dense and the font is tiny and hard to read which makes it very difficult to interpret. Perhaps consider breaking up this into multiple figures or moving some panels to supplementary information? Figures 3 and 6 are similarly difficult to absorb. Figure 4 would be easier to read if the axis labels were all larger size like 4H/4I.

To make interpretation clearer, we have significantly restructured our RNAseq figures, removing some analysis and moving other analysis (cytoscape) into supplemental data to allow for more space to increase both font and graph size.

Minor points -

**Fig 2A,B – ‘in terms of overall cortical gene expression’
– would be nice to have labels indicating units for scale bars**

Labels that indicated z-score and Euclidean distance are now included.

**Lines 452-454: ‘peak amplitude of the field excitatory postsynaptic potential (fEPSP) is shifted downward, indicating that LTP is significantly reduced compared to sham mice’
- phrasing here is unclear. Does ‘downward’ mean that the normalized amplitude is smaller than in controls after LTP induction, so that strength is potentiated to a lesser degree in HF-HI?**

We have changed our phrasing in the manuscript to make our meaning clearer.

- LTP is often quantified using the slope of the fEPSP. This probably gives a similar answer, but it could be nice to know if effects are comparable for comparison with other work.

The reviewer is right, slope is more widely used, but the use of peak amplitude is not unusual. The electrophysiologist authors on this paper have previously reported LTP and have used peak amplitude as their standard readout (e.g. Roseboom et al (2015), *Neurobiology of Learning and Memory* 125:265). When initially starting these experiments, we compared the slope and the peak amplitude in pilot experiments and they were similar, and we continued the remainder of the experiments with peak amplitude.

Fig. 4B – what were the peak amplitudes normalized to? Normalization is especially important for these input-output experiments and should be clearly stated.

The peak amplitude responses elicited at each stimulus intensity were normalized to the maximum response for each slice. This is now stated in the methods.

Fig. 4C – are the zoomed-in images stretched horizontally somehow? They seem elongated relative to the lower-mag images. Scale bars would be helpful here.

We have adjusted the image. The inner box itself represents 20uM, and this information is now included in the legend.

Fig. 4F – there is no statistical change in average paired-pulse ratio, but the variance of this measure does seem to increase. Could this be a real effect of injury, with different neurons affected in different directions?

We have added more n to the figure in response to Reviewer 2s comment, and included a PPR for the 1m timepoint. The variance between the sham and HF-HI groups is similar across both experiments, and there appears to be no effect of impact.

Fig S4C (mEPSC measurements) - these are an important counterpart to the PPR and silent synapse argument, but are harder to access in the supplemental materials. The data in S4A/B/C could be useful to show in the main figure panel especially since there is such a strong shift in mEPSC frequency.

We have included this information in the main figure panels for both the 1d and 1m timepoint.

**Line 475-476: ‘despite there being no difference between the sham and HF-HI groups in approximate rheobase following depolarizing current injections (Fig. S4D)’:
- is rheobase really identical? This seems somewhat surprising given the apparent difference in firing thresholds seen in the current-frequency plot in Fig 5B, and the examples in 5A/5C.**

The individual neuron rheobase data is available in Fig S5. The current frequency graphs show the frequency of firing, based on the average number of action potentials fired per group at each current step. The sham cells fire more action potentials at each step than the HF-HI cells, despite the fact that both cells fired at least one action potential at the same initial current step. Further, the measurement performed in this experiment is an approximation of the rheobase calculated from the delivery of 20pA current steps. Calculation of a more precise rheobase for each group would need to be performed using linear current ramps with smaller current amplitudes. It is possible that this may reveal a significant difference in rheobase between the two groups that was not observed in the average rheobase calculated with the present paradigm.

Line 476-477: This acute change in neuron excitability had normalized at 1m after HF-HI (Fig.5C-D).

- the groups are indistinguishable here in 5D, but the curves also look different than either group in 5C (that is they show a lower threshold). Can the authors comment on possible reasons for this?

There is no concrete explanation for the difference in threshold between the two time-points depicted in these graphs. In Fig S5 it is also clear that the rheobase is different for these two experimental timepoints. One can postulate that because the recordings for each time-point were separate experiments involving different sets of animals, and were recorded approximately 1 year apart, some variables may have been introduced that made the data produced at each time-point slightly different from one another. Also, the animals in the 1m group are older than the 1d mice, which could have had an effect on the cellular properties. The 1d animals were also only 24h after their last isoflurane exposure, whereas the 1m animals had a much longer time from isoflurane exposure.

To make sure our analysis was rigorous, we only included data from cells recorded from either group if there was a matched recording from a cell from the other group on the same day. This means that we

had within-day matches of sham and HFHI mice using all of the same perfusion and recording settings, solutions, and parameters.

Line 488-490: 'To determine whether changes in the relative expression or conductance of extrasynaptic NMDA receptors could be contributing to the functional changes that we observe' - what changes are the authors referring to? to the altered firing and reduced excitability described immediately beforehand? or the changes in plasticity earlier on? Both? The results jump from synaptic measurements, to firing properties, back to synaptic measurements. It might be easier to follow if the synaptic changes were all reported together – unless the authors are proposing that extrasynaptic NMDARs are responsible for both?

The sentence was meant to convey that we examined extrasynaptic receptor properties to determine whether changes in extrasynaptic NMDA receptor properties could be contributing to cognitive dysfunction, as an increase of conductance or relative expression of extrasynaptic receptors could indicate a potentially pro-apoptotic phenotype. This is not occurring. The paradigm we employed does not allow for the assessment of how these extrasynaptic receptors may be contributing directly to firing properties, excitability, or plasticity. It merely allows for a general determination of whether decay (and by extension, subunit composition or relative expression) may be altered.

We have revised this paragraph, the results section of the electrophysiology, and the figures to be clearer and kept the synaptic and firing properties better separated.

Line 498-499: 'Additionally, HF-HI does not appear to change glial glutamate transporter capabilities.'

- is this inferred from the NMDAR current decay? If so this should be stated, otherwise this statement implies transporter function were measured directly (e.g. with astrocyte recordings).

Yes, this is inferred and we have made this clearer in the discussion.

- 'with a net reduction in the AMPA/NMDA ratio which, along with changes in intrinsic excitability, drive acute and chronic plasticity impairments' – similar to above, these are likely to contribute but the causal relationship is not directly shown. The data here don't specifically establish whether plasticity effects arise from one or both of these changes. There could be contributions from other changes in synaptic protein content, signaling pathways, etc. Language should be more careful.

We have removed this sentence from the text.

Lines 346-349: In addition, Iba1, CD68 and GFAP staining reveal no changes in microglial or astrocyte activation/morphology in the cortex, hippocampus or corpus callosum of HF-HI mice compared to sham counterparts at 1d, 3d, or 7d post-HFHI (Fig. S1A,D-F) - lack of classical neuroinflammatory markers seems to be a major point of the study. The panels show single qualitative examples to support this idea. How was this evaluated or quantified for consistency? The legend refers to 'n = 6 per group' but it's not clear if this applies to both immunostaining and biochemistry?

Based on this comment and comments from other reviewers, we have limited our pathology results to 1d and 1m, and we have quantified the Iba1 and CD68 data from n = 4-8 animals per group per timepoint.

We have also included tau and Abeta analysis at 1d and 1m, cytokine assays at 1d and 1m, Iba1 and GFAP mRNA at 1d and 1m, and Iba1 Western blots at 1d and 1m.

Thank you to all the reviewers for the time and energy they have put into improving this manuscript. It is greatly appreciated by all of our team.

Reviewer #1 (Remarks to the Author):

This is a revised manuscript. The authors were attentive to the concerns brought forth in the initial review. These concerns have been addressed in the revision. More data have been provided and key points have been clarified. The presentation and interpretation of the data has been improved. Overall this is a novel and impactful study on neuropathologies associated with repeated closed head injuries.

Reviewer #3 (Remarks to the Author):

In this revised manuscript the authors have added substantial new electrophysiology data for better comparison of before and after injury that helps clarify the interpretation. The figures showing gene expression data are much easier to read and interpret. There are some remaining points about interpretation of findings and data that should be resolved. The main focus of the manuscript is synaptic adaptations, but it largely overlooks a considerable body of work examining changes in neuronal excitability and synaptic function for both mild injury and other models, and the findings should be framed and interpreted in the context of this previous work.

Major points:

The abstract states that 'HF-FI causes chronic adaptation of the AMPA-NMDA ratio in hippocampal neurons that underlie the changes to cognition.' The AMPA/NMDA deficits are found in hippocampus. The cognitive/behavioral deficits are consistent with hippocampus, but the gene expression data are from cortex and show widespread changes there too, and there are no electrophysiology data from brain areas besides hippocampus. While the data all support the general idea that their injury model causes mostly synaptic adaptations, there doesn't seem to be a good basis for this strong a claim attributing cognitive changes to one specific brain area.

The introduction refers to many studies on long-term pathology but mostly ignores a large body of work on changes in synaptic function and cellular excitability after injury. This includes among others a series of papers by Povlishock, Santhakumar et al 2000, 2001, and Goldstein et al 2012 / Tagge et al 2018. The last two are very relevant as they show related deficits in plasticity in hippocampus and prefrontal cortex for mild injury. Given that the synaptic effects are the main conclusion of this study, prior work in this area should be clearly and acknowledged in framing the experiments, in depth and not just in passing, and in discussing and interpreting results. Electrophysiological adaptation in more severe forms of injury also seems relevant.

Fig 2 - Intrinsic properties of CA1 neurons. If there are no changes in input resistance, how can we account for the strongly altered F-I curves? Was resting membrane potential measured? Rheobase is reported to be unchanged, but the curves in 2J look like they intersect zero at different points. Excitability is clearly different but these pieces of data don't all seem to fit together easily.

line 408 – 'To determine if these changes in synaptic profile could be attributed to an effect of neuroinflammation' - was neuroinflammation assayed in cortex where gene expression work was

performed? Some of the conclusions are based on combining information from the transcriptome dataset with the electrophysiology and neuropathology findings, but these seem to be mostly from different brain areas. The authors should be more careful about wording and drawing causal conclusions in these cases.

Line 665 – ‘first data to show that synaptic adaptation alone can cause strong behavioral effects...’ – the main phenotype is synaptic, and the authors assayed several common markers for neuropathology and found that it was minimal by these measures. That said, they haven’t done an exhaustive test ruling out all possible forms of pathology, and data show some microglial responses albeit limited. Transcriptome data are strongly biased towards but not completely limited to synaptic changes. This interpretation is broadly consistent with their findings but it seems difficult to conclusively rule out other factors.

Minor points:

In Fig 2K/L/M – ‘no change in lh’ – does this refer to no change in the magnitude of the currents? Or no change in their properties such as sag? The size of the currents and/or their density relative to membrane seems to be the most relevant but it’s not clear whether this is addressed.

In Figure 6G,H – the cumulative curves show quite large stepwise jumps that look different from the more typical plots shown for the earlier mEPSC data – what is the reason for this, how was the data quantified and binned?

line 444 – investigated basal strength of synapse using AMPA/NMDA ratio – is this really a measure of basal strength?

line 538 – Transcriptome data ‘points to synaptic modification rather than synaptic loss’. Do the data conclusively point towards altered synaptic function rather than synaptic loss? it seems possible that there could be some loss combined with adaptation in remaining synapses that might give broadly similar changes in gene expression. The spine data provide further evidence that spine loss didn’t occur, but this was measured for one specific class of dendritic branches in one cell type, and in hippocampus rather than cortex.

Thank you to the reviewers for their enthusiasm and excellent comments in response to our resubmission. The comments were constructive, and we have incorporated all the suggestions and feedback into this resubmission. The new text is marked with a line down the side of the page.

Reviewer 1

This is a revised manuscript. The authors were attentive to the concerns brought forth in the initial review. These concerns have been addressed in the revision. More data have been provided and key points have been clarified. The presentation and interpretation of the data has been improved. Overall this is a novel and impactful study on neuropathologies associated with repeated closed head injuries.

We greatly appreciate the feedback over the review cycle – it has made this manuscript stronger.

Reviewer 3

In this revised manuscript the authors have added substantial new electrophysiology data for better comparison of before and after injury that helps clarify the interpretation. The figures showing gene expression data are much easier to read and interpret. There are some remaining points about interpretation of findings and data that should be resolved. The main focus of the manuscript is synaptic adaptations, but it largely overlooks a considerable body of work examining changes in neuronal excitability and synaptic function for both mild injury and other models, and the findings should be framed and interpreted in the context of this previous work.

Major points:

The abstract states that ‘HF-FI causes chronic adaptation of the AMPA-NMDA ratio in hippocampal neurons that underlie the changes to cognition.’ The AMPA/NMDA deficits are found in hippocampus. The cognitive/behavioral deficits are consistent with hippocampus, but the gene expression data are from cortex and show widespread changes there too, and there are no electrophysiology data from brain areas besides hippocampus. While the data all support the general idea that their injury model causes mostly synaptic adaptations, there doesn’t seem to be a good basis for this strong a claim attributing cognitive changes to one specific brain area.

The reviewer is correct that the effect of HF-HI on synapses is not limited to the hippocampus, but is generalizable to the cortex and hippocampus (and possibly throughout the brain). We have removed the word hippocampus from the abstract.

For the resubmission, we have included new data from the cortex and hippocampus to strengthen the concept that the synaptic changes are widespread throughout the brain and occur in both the cortex and hippocampus. We now include dendritic spine counts from the dentate gyrus of the hippocampus, and also from Layer II/III of the cortex (Fig S5). We have also included RT-PCR data that confirms that key synaptic genes are also changed in the hippocampus (Fig S1).

The introduction refers to many studies on long-term pathology but mostly ignores a large body of work on changes in synaptic function and cellular excitability after injury. This includes among others a series of papers by Povlishock, Santhakumar et al 2000, 2001, and Goldstein et al 2012 / Tagge et al 2018. The last two are very relevant as they show related deficits in plasticity in

hippocampus and prefrontal cortex for mild injury. Given that the synaptic effects are the main conclusion of this study, prior work in this area should be clearly and acknowledged in framing the experiments, in depth and not just in passing, and in discussing and interpreting results. Electrophysiological adaptation in more severe forms of injury also seems relevant.

We have expanded our discussion about synaptic plasticity and cellular excitability and include more bodies of work from TBI. The following references are included in the discussion of plasticity in TBI:

- Aungst, S. L., Kabadi, S. V., Thompson, S. M., Stoica, B. A. & Faden, A. I. Repeated mild traumatic brain injury causes chronic neuroinflammation, changes in hippocampal synaptic plasticity, and associated cognitive deficits. *J Cereb Blood Flow Metab* **34**, 1223-1232, doi:10.1038/jcbfm.2014.75 (2014).
- Tagge, C. A. *et al.* Concussion, microvascular injury, and early tauopathy in young athletes after impact head injury and an impact concussion mouse model. *Brain* **141**, 422-458, doi:10.1093/brain/awx350 (2018).
- Miyazaki, S. *et al.* Enduring suppression of hippocampal long-term potentiation following traumatic brain injury in rat. *Brain Res* **585**, 335-339, doi:10.1016/0006-8993(92)91232-4 (1992).
- Albensi, B. C., Sullivan, P. G., Thompson, M. B., Scheff, S. W. & Mattson, M. P. Cyclosporin ameliorates traumatic brain-injury-induced alterations of hippocampal synaptic plasticity. *Exp Neurol* **162**, 385-389, doi:10.1006/exnr.1999.7338 (2000).
- Goldstein, L. E. *et al.* Chronic traumatic encephalopathy in blast-exposed military veterans and a blast neurotrauma mouse model. *Sci Transl Med* **4**, 134ra160, doi:10.1126/scitranslmed.3003716 (2012).
- Mei, Z. *et al.* Memantine improves outcomes after repetitive traumatic brain injury. *Behav Brain Res*, doi:10.1016/j.bbr.2017.04.017 (2017).

The following refernces are included in the discussion of neuron excitability and intrinsic properties in TBI:

- Hanell, A., Greer, J. E. & Jacobs, K. M. Increased Network Excitability Due to Altered Synaptic Inputs to Neocortical Layer V Intact and Axotomized Pyramidal Neurons after Mild Traumatic Brain Injury. *J Neurotrauma* **32**, 1590-1598, doi:10.1089/neu.2014.3592 (2015).
- Bugay, V. *et al.* A Mouse Model of Repetitive Blast Traumatic Brain Injury Reveals Post-Trauma Seizures and Increased Neuronal Excitability. *J Neurotrauma* **37**, 248-261, doi:10.1089/neu.2018.6333 (2020).
- Karimi, S. A., Hosseinmardi, N., Sayyah, M., Hajisoltani, R. & Janahmadi, M. Enhancement of intrinsic neuronal excitability-mediated by a reduction in hyperpolarization-activated cation current (I_h) in hippocampal CA1 neurons in a rat model of traumatic brain injury. *Hippocampus* **31**, 156-169, doi:10.1002/hipo.23270 (2021).
- Nichols, J., Perez, R., Wu, C., Adelson, P. D. & Anderson, T. Traumatic brain injury induces rapid enhancement of cortical excitability in juvenile rats. *CNS Neurosci Ther* **21**, 193-203, doi:10.1111/cns.12351 (2015).
- Santhakumar, V., Ratzliff, A. D., Jeng, J., Toth, Z. & Soltesz, I. Long-term hyperexcitability in the hippocampus after experimental head trauma. *Ann Neurol* **50**, 708-717, doi:10.1002/ana.1230 (2001).
- Greer, J. E., Povlishock, J. T. & Jacobs, K. M. Electrophysiological abnormalities in both axotomized and nonaxotomized pyramidal neurons following mild traumatic brain injury. *J Neurosci* **32**, 6682-6687, doi:10.1523/JNEUROSCI.0881-12.2012 (2012).

Deng, P. & Xu, Z. C. Contribution of Ih to neuronal damage in the hippocampus after traumatic brain injury in rats. *J Neurotrauma* **28**, 1173-1183, doi:10.1089/neu.2010.1683 (2011).

Fig 2 - Intrinsic properties of CA1 neurons. If there are no changes in input resistance, how can we account for the strongly altered F-I curves? Was resting membrane potential measured? Rheobase is reported to be unchanged, but the curves in 2J look like they intersect zero at different points. Excitability is clearly different but these pieces of data don't all seem to fit together easily.

We thank the reviewer for their observation on this point. We went back to our source data to understand the disconnect. When initially setting up these intrinsic property experiments we used larger current steps than we used in our final F-I curves in Fig 2 and Fig 6 - but some of this data were included in the rheobase analysis. We now only present the data from the pair-matched sham and HFHI slices used in the F-I curves at 1d and 1m (n=1 for each group collected on each experimental day with 20mA steps, total n=8/group). As the reviewer has noted, there is an increase in the rheobase in HFHI cells at the 1d timepoint, and a strong trend to a decreased input resistance. We adjusted the panels in Fig 2 to only reflect cells run under the same experimental conditions, and we confirmed that our 1m intrinsic data only includes cells run under identical parameters.

line 408 – ‘To determine if these changes in synaptic profile could be attributed to an effect of neuroinflammation’ - was neuroinflammation assayed in cortex where gene expression work was performed? Some of the conclusions are based on combining information from the transcriptome dataset with the electrophysiology and neuropathology findings, but these seem to be mostly from different brain areas. The authors should be more careful about wording and drawing causal conclusions in these cases.

We have changed the wording in this line to “To study neuroinflammation in HF-HI mice, we analyzed Iba1 and CD68 immunohistochemistry in the mouse brain”.

To address the disconnect between brain regions, we have included new PCR data that shows that we also have a change in synaptic mRNA profiles in the hippocampus (Fig S1). For the cortex, we have cortical IL1b and TNFa cytokine ELISAs at both 24h and 1m (Fig S3) to show that these cytokine signals are not elevated in the cortex. In our original submission, we had also included these ELISAs at 6h from cortex and hippocampus (there were no changes), but to simplify our manuscript we removed any timepoints outside of 1d and 1m from the manuscript. We also do not detect changes in *Aif1* (microglia) and *Gfap* (astrocytes) mRNA in the cortex (Fig S3). There are not CD68-positive cells present in the cortex of sham or HFHI mice at 1d or 1m mice to quantify, demonstrating a lack of activated microglia in the cortex at both timepoints.

Line 665 – ‘first data to show that synaptic adaptation alone can cause strong behavioral effects...’ – the main phenotype is synaptic, and the authors assayed several common markers for neuropathology and found that it was minimal by these measures. That said, they haven't done an exhaustive test ruling out all possible forms of pathology, and data show some microglial responses albeit limited. Transcriptome data are strongly biased towards but not completely limited to synaptic changes. This interpretation is broadly consistent with their findings but it seems difficult to conclusively rule out other factors.

It is difficult to include all the negative data we have for TBI and AD pathology in HF-HI mice (we simplified our pathology figure for the initial resubmission), but we respect the point the reviewer is

making. We have changed this statement to say “These data are consistent with synaptic adaptation causing strong cognitive deficits following head impact”

Minor points:

In Fig 2K/L/M – ‘no change in lh’ – does this refer to no change in the magnitude of the currents? Or no change in their properties such as sag? The size of the currents and/or their density relative to membrane seems to be the most relevant but it’s not clear whether this is addressed.

We measured sag in our work, and we have changed the wording to better reflect the experiments we conducted.

In Figure 6G,H – the cumulative curves show quite large stepwise jumps that look different from the more typical plots shown for the earlier mEPSC data – what is the reason for this, how was the data quantified and binned?

New curves have been produced from the raw data.

line 444 – investigated basal strength of synapse using AMPA/NMDA ratio – is this really a measure of basal strength?

We have adjusted the sentence to remove this phrasing.

line 538 – Transcriptome data ‘points to synaptic modification rather than synaptic loss’. Do the data conclusively point towards altered synaptic function rather than synaptic loss? it seems possible that there could be some loss combined with adaptation in remaining synapses that might give broadly similar changes in gene expression. The spine data provide further evidence that spine loss didn’t occur, but this was measured for one specific class of dendritic branches in one cell type, and in hippocampus rather than cortex.

We have examined spine density in many different brain regions in this model. We have included new data on the spine density of Layers II/III of the cortex (apical obliques and basal shaft dendrites), pyramidal neurons of the CA1 and granule cells of the dentate gyrus of the hippocampus in these mice (Fig S5). These data show that spine density is not changed in HF-HI mice compared to sham mice.

Thank you for your constructive comments and attention to detail. It has helped improve our manuscript and the comments are greatly appreciated by all of our team.

Reviewer #3 (Remarks to the Author):

The authors have done a further major round of including additional data, as well as re-analyzing existing datasets and calibrating some specific interpretations. I very much appreciate that they have been very responsive to addressing all the points raised by my review in a substantive way. This integrative study will be a nice addition to the literature on mild injury. I have no further concerns or comments.